# PEGylated Strontium Sulfite Nanoparticles with Spontaneously Formed Surface-Embedded Protein Corona Restrict Off-Target Distribution and Accelerate Breast Tumour-Selective Delivery of siRNA

**DOI:** 10.3390/jfb13040211

**Published:** 2022-11-01

**Authors:** Md. Emranul Karim, Ezharul Hoque Chowdhury

**Affiliations:** Jeffrey Cheah School of Medicine and Health Sciences, Monash University Malaysia, Jalan Lagoon Selatan, Bandar Sunway, Petaling Jaya 47500, Malaysia

**Keywords:** EGFR, inorganic nanoparticles, strontium sulfite nanoparticles, protein corona, mass spectrometry, biodistribution, breast tumour regression

## Abstract

As transporters of RNAi therapeutics in preclinical and clinical studies, the application of nanoparticles is often hindered by their susceptibility to opsonin-mediated clearance, poor biological stability, ineffectual targeting, and undesirable effects on healthy cells. Prolonging the blood circulation time while minimizing the off-target distribution and associated toxicity is indispensable for the establishment of a clinically viable delivery system for therapeutic small interfering RNAs (siRNAs). Herein, we report a scalable and straightforward approach to fabricate non-toxic and biodegradable pH-responsive strontium sulfite nanoparticles (SSNs) wrapped with a hydrophilic coating material, biotinylated PEG to lessen unforeseen biological interactions. Surface functionalization of SSNs with PEG led to the generation of small and uniformly distributed particles with a significant affinity towards siRNAs and augmented internalization into breast cancer cells. A triple quadrupole liquid chromatography-mass spectrometry (LC-MS) was deployed to identify the proteins entrapped onto the SSNs, with the help of SwissProt.Mus_musculus database. The results demonstrated the reduction of opsonin proteins adsorption owing to the stealth effect of PEG. The distribution of PEGylated SSNs in mice after 4 h and 24 h of intravenous administration in breast tumour-bearing mice was found to be significantly less to the organs of the reticuloendothelial system (RES) and augmented accumulation in the tumour region. The anti-EGFR siRNA-loaded PEG-SSNs exerted a significant inhibitory effect on tumour development in the murine breast cancer model without any significant toxicity to healthy tissues. Therefore, PEGylated SSNs open up a new avenue for tumour-selective efficient delivery of siRNAs in managing breast cancer.

## 1. Introduction

The biotechnology revolution has brought forward RNA interference (RNAi) as a potential therapeutic option over classical chemotherapy in cancer management [1]. Short interfering RNA (siRNA) is an efficient approach to silence the expression of endogenous genes responsible for cancer development [2]. However, the delivery of naked or modified siRNA into biological systems is quite challenging. The naked or unmodified siRNA is associated with a short half-life due to its susceptibility to nuclease-mediated degradation in the systemic circulation, rapid renal clearance, poor cellular internalization rate, and potential off-target effects [3,4,5,6,7]. Aiming to surmount these barriers, a lot of synthetic transporters have been designed to ferry the siRNA into the cytoplasm of target tumour cells [4]. Among them, organic and inorganic nanoparticles (NPs) have been widely explored, especially in cancer gene therapy [8,9]. Compared with organic counterparts, inorganic NPs possess some unique physicochemical characteristics with regard to synthesis and surface-functionalization strategies, and the resultant effects on their size, shape, electrical, and optical properties [10,11].

A significant number of preclinical studies and clinical studies on cancer nanotherapeutics have been reported in the last few years, revealing the challenges and hurdles of the safe and effective delivery of potential therapeutic NPs in cancer patients [12]. It is hypothesized that most of the intravenously administered NPs that accumulate in the target tumour cells extravasate from the leaky tumour vasculature through enhanced permeability and retention (EPR) effects or by active targeting [13,14,15,16,17]. However, during traveling from the site of administration to the site of action, NPs encounter extracellular components of biological systems, that determine the fate and navigating pattern of injected NPs inside the body [18]. In principle, when NPs come into contact with biological fluids, a variety of biomolecules, such as proteins, lipids, and carbohydrates from blood and extracellular matrix (ECM) are sequestered with NPs owing to their large surface-to-volume ratio and surface charge [19,20]. The interactions between NPs and proteins form a crown or layer around the surface, called ‘protein corona’ (PC), transforming the bare NPs into a new biological entity “NPs-PC” complex [19,21,22,23]. The newly formed NPs-PC complex alters the physicochemical properties, tissue distribution, cellular internalization, and toxicity of the NPs, determining the pharmacokinetics and pharmacodynamics of the latter [20,24,25].

Depending on the physicochemical properties and exposure time of NPs for exchanging proteins, PC might be hard or soft [26]. Hard corona comprises the high-affinity proteins that form an irreversible and tight layer on the surface of NPs and can be exchangeable for many hours. On the other hand, a soft corona consists of low-affinity proteins, forming a reversible and cleavable layer on the interface of NPs and being exchangeable rapidly within minutes or seconds [22,27,28]. The NPs-PC complex contains hundreds of different proteins; the most abundant identified proteins are apolipoproteins, adhesion mediators, signaling and transport proteins, immunoglobulin (IgG), complement components, and coagulation factors [29,30]. Among them, complement and IgG proteins play a vital role in the opsonization of NPs, and the opsonized NPs are readily taken by macrophages, including dendritic cells and blood monocytes present in the liver, spleen, and lymph nodes [25,31]. Consequently, complexes of proteins/NPs are subjected to the opsonin-mediated uptake by macrophage of the reticuloendothelial system (RES), non-specific biodistribution, rapid clearance, and formation of thrombosis, declining therapeutic efficiency of drug-loaded NPs [32]. Conversely, serum albumin and apolipoprotein, known as dysopsonins which are also found in protein corona, prevent the opsonin-mediated macrophage clearance and increase the stability of NPs in the biological fluid [33,34,35]. As the inorganic NPs contain charges on their surface, after entering the blood circulation, they are prone to interactions with serums—proteins that adsorb onto the surface of NPs, being readily taken up by the macrophages [33,36]. Moreover, the small-sized NPs offer a large surface area for adsorbing blood proteins [19,37,38]. From the pharmaceutical point of view, premature degradation before reaching the target site results in the drug’s sub-therapeutic action. So, it is necessary to minimize the unwanted clearance of NPs for maximizing their therapeutic potential and targetability, while reducing the off-target effects [39].

A surface modification would be a potential approach to avoid opsonization and systemic instability of NPs for improved therapeutic action with reduced unwanted effects of nanotherapeutics. Polyethylene glycol (PEG), a water-soluble, non-ionic, and biocompatible coating material is widely used to extend the blood circulation time of drug-loaded NPs and thus enhance therapeutic efficacy [40,41,42,43,44]. It helps to provide a stealth envelop onto the surface of NPs and aids to keep the NPs in a colloidal state, thereby preventing particle agglomeration [45]. PEGylation is the most acceptable way to reduce the protein adsorption onto the surface of NPs, thus inhibiting macrophage uptake via opsonin receptors and extending the half-life of the NPs in blood [46]. Moreover, grafting of NPs surfaces with PEG alters the physicochemical properties of NPs which might be helpful in the formation of NPs with favorable pharmacokinetics profiles [47,48,49,50]. Apart from PEG, other polymers like poly (vinylpyrrolidone) (PVP), dextran, chitosan, pullulan, etc. have been reported to reduce the protein adsorption onto the surface of NPs. The addition of polymers that are responsive to pH might help to release loaded siRNAs or drugs, escaping the protein entrapment from systemic circulation [45,51].

The epidermal growth factor receptor (EGFR)) is a receptor tyrosine kinase of the ErbB/HER family, that forms homo- and hetero-dimer receptors and modulates cancer cell proliferation, cellular adhesion, motility, apoptosis, progression, and migration by activating signal transduction pathways [52,53,54,55]. EGFR overexpression has been reported for all types of breast cancers, particularly triple-negative breast cancer (TNBC) and inflammatory breast cancer (IBC) [56,57,58]. The overexpression of EGFRs in breast cancer, their aggressive behaviors, and poor clinical outcomes make them ideal targets for cancer gene therapy. Silencing of EGFR via anti-EGFR siRNA could be an effective strategy for cancer management.

We previously reported the fabrication of pH-responsive strontium sulfite nanoparticles (SSNs) and the application of the nanocarrier to efficaciously transport ROS1 siRNA into tumour cells, demonstrating a significant reduction in tumour volume in a murine breast cancer model [58]. To make the system more effective for systemic administration, we modified the surface of SSNs with biotin-PEG for improving their blood circulation time, minimizing off-target distribution, and augmenting tumour accumulation. The association of biotin-PEG with NPs resulted in the reduction of the hybrid particle size by inhibiting particle aggregation. The size, morphology, elemental analysis, and stability study of PEG-SSNs supported our original assumption that the small-sized PEG-SSNs could protect the siRNA from biological instability and increase the accumulation in the intended tumour site via the EPR effect. We also examined the binding affinity of PEG-SSNs to siRNA and their capability to carry siRNA into the tumour cell and silence the overexpressed EGFR gene with anti-EGFR siRNA in human and murine breast cancer cell lines. Next, the quantification of the proteins entrapped onto the surface of SSNs and PEG-SSNs was analyzed using triple quadrupole liquid chromatography-mass spectroscopy (QQQ LC/MS) and attempted to correlate with the time-dependent distribution patterns of intravenously administered fluorescence siRNA-loaded SSNs and PEG-SSNs in major organs and tumours. Finally, the tumour regression capacity of SSNs and PEG-SSNs with loaded anti-EGFR siRNA and the safety profile of the formulations were evaluated in the same breast tumour-bearing mouse model.

## 2. Materials and Methods

### 2.1. Materials

Strontium chloride (SrCl_2_), sodium sulfite (Na_2_SO_3_), and 4-(2-hydroxyethyl)-1-piperazineethanesulfonic acid (HEPES) were obtained from Sigma-Aldrich (St Louis, MO, USA). Dulbecco’s modified eagle medium (DMEM), dimethyl sulphoxide (DMSO), thiazolyl blue tetrazolium bromide (MTT), and ethylene diamine tetra acetic acid (EDTA) were purchased from Sigma-Aldrich (St. Louis, MO, USA). DMEM powder, fetal bovine serum (FBS), trypsin-ethylene diamine tetra acetate (trypsin-EDTA), and penicillin-streptomycin were from Gibco BRL (CA). Sigma Aldrich provided the poly (ethylene glycol) 2-aminoethyl ether biotin (St. Louis, MO, USA). The company Qiagen (Hilden, Germany) provided the RNase-free water, which was used to dissolve all of the siRNAs employed in this investigation. MCF-7 and 4T1 cells were originally obtained from ATCC (Manassas, VA, USA).

### 2.2. Synthesis of SSNs and PEG-SSNs

SSNs were synthesized by mixing 40 mM of SrCl_2_ and 10 mM of Na_2_SO_3_ in 50 µL of an aqueous solution, incubating at 37 °C for 30 min, the mixture was then topped up with 1 mL of DMEM medium (pH 7.5). For PEG-SSNs, 40 mM of SrCl_2_ and 10 mM of Na_2_SO_3_ were added in 50 µL of an aqueous solution, and incubated for 30 min at 37 °C. After the initial incubation, various concentrations (1, 5, 10, 15 µL) of biotin-PEG (1 µM) were added before being incubated again for 10 min at 37 °C.

### 2.3. Turbidity Measurement and Optical Imaging of PEG-SSNs

SSNs and PEG-SSNs were prepared as mentioned above. At 320 nm, a spectrophotometric analysis was done to measure the absorbance (turbidity) patterns of SSNs and PEG-SSNs formed at different concentrations of biotin-PEG (1, 5, 10, 15 µL of 1 µM) by using a Jasco UV-VIS spectrophotometer (Oklahoma City, OK, USA). Microscopic observation of formed PEG-SSNs nanoparticles was done using an Olympus Microscope CKX41 in the bright field with 10× magnification. The data were graphed with mean SD, and each experiment was carried out in triplicate at room temperature.

### 2.4. Characterizations of PEG-SSNs by DLS

SSNs and PEG-SSNs were made by combining 40 mM of SrCl_2_ and 10 mM of Na_2_SO_3_ in 50 µL of an aqueous solution, then incubating the mixture at 37 °C for 30 min and, for the synthesis of PEG-SSNs, 1 µL of Biotin PEG (1 µM) were added to SSNs solutions and incubated for 10 min at 37 °C. Malvern nano zeta sizer (Worcestershire, UK) and accompanying software were used to measure the particle size, surface charge, particle size distribution, and PDI value. All samples were analyzed and a graph with mean ± SD was created.

### 2.5. Field Emission Scanning Electron Microscopy (FE-SEM) of PEG-SSNs

The actual size, surface, and shape of SSNs and PEG-SSNs were observed by using FE-SEM. PEG-SSNs were synthesized by mixing 40 mM of SrCl_2_, 10 mM of Na_2_SO_3_, and 1 µL of 1 µM biotin PEG, where SSNs were fabricated without PEG by using the same amount of SrCl_2_ and Na_2_SO_3_. The synthesized particles were then centrifuged for 15 min at 13,000 rpm. The resulting pellet was suspended in 500 µL of water. After that, 3 µL of the particle suspension was put on a glass slide to dry at 37 °C for an hour. The dried sample was placed into a sample holder coated with carbon tape and subjected to platinum sputtering for 40 s with 30 mA sputter current at 2.30 tooling factor. FE-SEM (Hitachi/SU8010, Tokyo, Japan) was used to observe the final sputtered particles at 5.00 kV.

### 2.6. Energy Dispersive X-ray (EDX) Analysis of PEG-SSNs

EDX was used to determine the elemental composition of SSNs and PEG-SSNs. The preparation of SSNs involved adding 40 mM of SrCl_2_ and 10 mM of Na_2_SO_3_ and 1 µL of Biotin PEG (1 µM) were added to prepare PEG-SSNs with the same concentrations of SrCl_2_ and Na_2_SO_3_. After 15 min of centrifuging the particles at 13,000× *g* rpm, 3 µL of the particle suspension was transferred to a glass slide and dried for an hour at 37 °C. The dried sample was then placed onto a carbon tape-coated sample holder, and subjected to platinum sputtering with 30 mA sputter current at 2.30 tooling factor for 40 s. Next, the sputtered particles were observed at 5.00 kV using FE-SEM (Hitachi/SU8010, Tokyo, Japan), and an EDX analyzer was used to determine the elements present (X-maX, 50 mm^2^ HORIBA, Kyoto, Japan).

### 2.7. Stability of PEG-SSNs in Mice Plasma

Malvern nano zeta sizer (Worcestershire, UK) was used to evaluate the stability and size distribution pattern. Fresh blood from female Balb/c mice was drawn into centrifuge tubes containing Heparin, and the plasma was obtained by centrifuging the tubes at 1000× *g* RCF for 10 min at 4 °C. The supernatant was collected and further centrifuged for 15 min at 2000× *g* RCF at 4 °C for separating plasma and blood cells. PEG-SSNs were synthesized by using 40 mM of SrCl_2_, 10 mM of Na_2_SO_3_, and 1 µL of 1 µM biotin PEG, dispersed in deionized water, DMEM cell culture medium, and mice plasma (10% in PBS) and subjected to DLS measurement.

### 2.8. siRNA Binding Affinity of PEG-SSNs

AF 488 Allstars negative siRNA was dissolved in 200 µL of DMEM at various concentrations (0, 2, 4, 6, 8, 10 nM), and the fluorescence intensity was evaluated with a λex = 490 nm and λem = 535 nm by using a 2030 multilabel reader victor TM X5 (Perkin Elmer, MA, CA, USA). The Perkin Elmer 2030 management software was used to examine the data. A standard curve was produced by graphing the fluorescence intensity values against the siRNA concentrations after each sample was evaluated in triplicate. The quantity of siRNA bound to the NPs was then determined using the standard curve. In order to create an NPs-siRNA complex, SSNs and PEG-SSNs were generated (as mentioned above by adding 40 mM of SrCl_2_ and10 mM of Na_2_SO_3_ with and without 1 µL of 1 µM biotin PEG) and complexed with 10 nM of AF 488 siRNA. The supernatants were collected after the samples were centrifuged at 13,000× *g* rpm for 15 min at 4 °C, and the fluorescence intensity was assessed. The following formula was used to determine the binding affinity of siRNA to variously formed NPs:% of siRNA binding = (X_initial-X_free)/X_initial × 100%
where X free represents the siRNA concentrations in the supernatant after centrifuging NPs-siRNA complexes, and X initial represents the total concentration of siRNA utilised in the experiment, which was 10 nM. The samples were created in triplicate and displayed as mean SD.

### 2.9. Biocompatibility Assay of PEG-SSNs

Hemolysis assay. Fresh blood was collected from female Balb/c mice and centrifuged at 2000× *g* rpm for 10 min at 4 **°**C for separating red blood cells (RBCs) from the blood. The collected RBCs were then washed with PBS three times and diluted with 20% (*v*/*v*) of PBS. The PEG-SSNs were prepared by adding 40 mM of SrCl_2_ and 10 mM of Na_2_SO_3_ and 1 µL of 1 M biotin PEG and added to RBCs solution before incubation at 37 °C for 60 min. The samples were then centrifuged for 5 min at 4000× *g* rpm, and the supernatant was examined using a UV-VIS spectrophotometer with a 541 nm wavelength at a wavelength of 541 nm. Deionized water and PBS were taken as positive and negative controls, respectively. The hemolysis percentage of PEG-SSNs was calculated by using the following equation:% of Hemolysis = ([Abs]_sample − [Abs[_Neg])/([Abs]_Pos − [Abs]_Neg)(1)

Cytotoxicity assay. Approximately 50,000 MCF-7,4T1 and HEK 293 cells were seeded on a 24-well plate and incubated overnight. The following day, cells were treated with SSNs and PEG-SSNs and incubated at 37 °C and 5% CO_2_ for 48 h. Next, 50 µL of MTT (5 mg/mL in PBS) solution was added to each well and the plate with MTT-treated cells was incubated for 4 h. Next, 300 µL of DMSO (dimethyl sulfoxide) was added to dissolve the resultant formazan. The absorbance was analyzed on a microplate reader (Biorad, Hercules, CA, USA) at 595 nm wavelength with reference to 630 nm.

### 2.10. Evaluation of Cellular Internalization of PEG-SSNs

MCF-7 cells were seeded on coverslips in a 6-well plate at a seeding density of 5 × 10^4^ cells per well. The next day, the cells were treated with free AF-488 labeled siRNA, and SSNs and PEG-SSNs with loaded fluorescent siRNA. After 1 h of incubation, cells treated with media as control, free fluorescence siRNA and fluorescence siRNA tagged-SSNs and -PEG-SSNs were washed three times with PBS, followed by fixation for 15 min at 37 °C with 4% paraformaldehyde solution in PBS. Then the cells were further treated with DAPI for 15 min after washing three times with PBS. Finally, all the images were taken using a Nikon Confocal C1 laser scanning microscope (Nikon Instruments, Tokyo, Japan).

### 2.11. Release Profile of SSNs and PEG-SSNs Complexes

The siRNA-loaded SSNs with or without PEG were formulated using 40 mM of SrCl_2_, 10 mM of Na_2_SO_3_, and 1 µL of 1 M biotin PEG in 50 µL of aqueous solution. After incubation at 37 ◦C for 30 min, SSNs/PEG-SSNs/siRNA complexes were subsequently suspended into 950 µL DMEM media of different pHs (7.5, 7.0, 6.5, 6.0, 5.5, and 5.0). Afterward, the particle suspensions were centrifuged at 6000× *g* rpm for 20 min at 4 °C with a refrigerated bench-top microcentrifuge. The supernatant was discarded and the pellet was mixed with 10 mM EDTA in PBS to dissolve the particles and release the bound siRNAs, prior to fluorescence intensity measurement with an excitation wavelength of 490 nm and an emission wavelength of 535 nm using PerkinElmer 2030 manager software attached with 2030 multilabel reader victor X5 (PerkinElmer, Waltham, MA, USA). The concentrations of AF-488 siRNA present in the suspension were calculated from the fluorescence intensity, using the standard curve. Data were represented as the % of siRNA release from NPs. The % siRNA release profile was calculated using the following formula:% of siRNA release from NPs = ([X] NP bound siRNA at pH 7.5 − [Y] NP bound siRNA at different pH)/[X] NP bound siRNA at pH 7.5× 100%
where “[Y] NP bound siRNA at different pH” is the fluorescence intensity of siRNA bound with NPs at pH 7.0, 6.5, 6.0, 5.5, and 5.0 and “[X] NP bound siRNA at pH 7.5” is the fluorescence intensity of AF-488 siRNA bound with NPs at pH 7.5. Data were represented as mean ± SD for triplicates.

### 2.12. In Vitro Cytotoxicity Assay of SSNs and PEG-SSNs Complexed with EGFR siRNA

MCF-7 and 4T1 cells were cultured in DMEM supplemented with 10% FBS and 1% penicillin/streptomycin at 37 °C in a humidified 5% CO_2_ chamber and approximately 50,000 cells/well were seeded in 24-well plates. After the overnight incubation, the cells were treated with SSNs, PEG-SSNs, SSNs-EGFR, and PEG-SSNs-EGFR and incubated for 48 h. SSNs were prepared by adding 40 mM of SrCl_2_ and 10 mM of Na_2_SO_3_ while 1 µL of 1 µM biotin PEG was additionally added to make PEG-SSNs. Next, 1 nM of EGFR siRNA was added after mixing the salts (SrCl_2_ and Na_2_SO_3_) for making SSNs-EGFR and PEG-SSNs-EGFR complexes. MTT assay was performed to determine cytotoxicity after 48 h of incubation. Each well received 50 µL of the MTT solution, which was then incubated for 4 h. The formazan crystals were mixed with 300 µL of DMSO before the absorbance was measured using a microplate reader (Biorad, Hercules, CA, USA) at 595 nm and 630 nm (reference) wavelengths. The following equation was used to determine the % of cell viability in treated samples:% cell viability = (Absorbance of the treated sample)/(Absorbance of control) × 100%(2)

All tests were performed in triplicate, and the data were presented as a percentage of cell viability with mean ± SD.

### 2.13. EGFR Expression Study by RT–PCR

After being seeded in 24-well plates with 5 × 10^4^ MCF-7 cells per well, the cells were treated with SSNs, PEG-SSNs, EGFR-SSN, and EGFR-PEG-SSNs (1 nM of EGFR siRNA). The cells were subsequently incubated at 37 °C for 48 h and total RNA was extracted from the cells using innuPREP RNA Mini Kit (Analytikjena, Berlin, Germany) according to the manufacturer’s protocols. The Nano Drop Spectrophotometer was used to measure the quantity and quality of RNA at a ratio of 260/280. Reverse transcription was conducted in RNAses-free PCR tubes using SensiFAST cDNA Synthesis Kit (Biolin, UK). Quantitative Real Time–PCR was done by SensiFAST SYBR Hi-ROX master mix (Biolin, UK). The housekeeping gene β-actin was used to normalise the EGFR gene expression level using the 2(−∆∆CT) method. Primer pair sequences: EGFR, forward primer: GCGTTCGGCACGGTGTATAA, reverse primer: GGCTTTCGGAGATGTTGCTTC; β-actin, forward primer: ACACTGTGCCCATCTACGAG reverse primer: TCAACGTCACACTTCATGATG.

### 2.14. In-Solution Digestion of Protein Corona Formed on SSNs and PEG-SSNs for Mass Spectrometric Analysis

SSNs and PEG-SSNs fabricated (as discussed above using 40 mM of SrCl_2_ and 10 mM of Na_2_SO_3_ in the absence or presence of 1 µL of 1 M biotin-PEG) were subjected to incubation for 15 min at 37 °C with mouse plasma (10%). After centrifugation of the particle suspensions at 13,000× *g* rpm for 15 min, the supernatants were removed, and the pellets were collected. Next, 100 µL of 50 mM EDTA in water was used to dissolve the pellets. The protein mixture (released from pellets) was mixed with 25 µL of 100 mM ammonium bicarbonate solution, 25 µL of tetrafluoroethylene (TFE) denaturing agent, and 1 µL of dithiothreitol (DTT) (200 mM) solution before being heated at 60 °C for an hour under a heating block. Next, 4 µL of 200 mM iodoacetamide (IAM) were added, vortexed and then the protein mixture (representing protein corona) was incubated in the dark at room temperature for 1 h. Next, 1 µL of 200 mM DTT solution was added to the protein mixture and allowed to incubate at room temperature for 1 h in the dark. Afterward, the treated protein mixture was then incubated for 4 to 18 h at room temperature with 100 µL of ammonium bicarbonate solution (100 mM) and 25 µL of MS Grade trypsin (1 µg/mL). After adding 1 µL of formic acid to stop the reaction, the samples underwent an overnight speed vacuum before being subjected to Q-TOF LC-MS/MS analysis.

### 2.15. Sample Preparation for Mass Spectrometry-Based Proteomics

The dried peptide digest was dissolved in 10 µL of formic acid (0.1%) in water. Following that, samples were sonicated in an ultrasonic water bath for 10 min with ice to maintain a low temperature. Samples were centrifuged (14,000× *g*, 5 min) before being immediately transferred to an LC-QTOF auto-sampler for analysis using a volume of 5 µL of the supernatant. Using a previously described procedure with minimal modifications, high-efficiency nanoflow liquid chromatography electrospray-ionization in combination with mass spectrometry, protein identification, and quantification by automated de novo sequencing (PEAKS Studio 8.5), was carried out [45].

### 2.16. Tumour Induction in Mice

Female nude Balb/c mice aged 6 to 8 weeks, weighing 20 to 25 g, were purchased from the Animal Facility Unit at the School of Medicine and Health Science at Monash University. They were kept in 12:12 light:dark conditions and given access to food and water as needed. All research was conducted in accordance with a protocol that was approved by the MONASH Animal Ethics Committee (MARP/2016/126). Approximately 1 × 10^5^ of 4T1 cells were suspended in 100 µL of PBS and administered subcutaneously to the mammary pad of mice.

### 2.17. Biodistribution Study of SSNs and PEG-SSNs Carrying Fluorescent siRNA

When the tumour volume reached 75 mm^3^ on day 15 of injection, the tumour-bearing mice were treated with SSNs and PEG-SSNs (prepared with 1 and 10 µL of 1 M biotin-PEG) carrying fluorescent AF-488 labeled neg. siRNA, through tail vein injection. Next, 75 nM of AF-488 siRNA was added along with 40 mM of SrCl_2_ and 10 mM of Na_2_SO_3_, and the resultant mixture was subjected to incubation for 30 min at 37 °C to prepare siRNA/SSNs, which were subsequently mixed with 1 µL and 10 µL of 1 M biotin-PEG to generate siRNA/PEGylated SSNs.

After 4 or 24 h of treatment, mice were euthanised humanely by cervical dislocation. Major organs such as the lung, brain, liver, kidney, spleen, heart, and tumour were then collected and washed two times in cold PBS before being mixed with 500 µL of lysis buffer per 500 mg of tissue mass. Using a mechanical homogenizer, tissues from various organs were lysed while being kept on ice until a thoroughly homogenized solution was formed. For measuring the fluorescence intensity of AF-488 labelled siRNA using a 2030 multilabel reader Victor TM X5 (Perkin Elmer) at ex = 490 nm and em = 535 nm, the solutions of tissue lysates were centrifuged for 25 min at 4 °C with 8000× *g* rpm. Next, 150 µL of the supernatant was then added to each well of a black 96-well. Following blank-correction using a set of mice that had not received any treatment, the data were shown as mean SEM of fluorescence intensity/500 mg of tissue mass for each tissue.

### 2.18. In vivo Antitumour Efficacy of EGFR siRNA-Loaded SSNs and PEG-SSNs

After the tumour reached an average volume of ~22 mm^3^ at around day 10–11 of subcutaneous injection, tumour-bearing mice were injected with free SSNs, free PEG-SSNs, EGFR-SSNs (50 nM of EGFR siRNA), and EGFR-PEG-SSNs (50 nM of EGFR siRNA) intravenously (tail-vein) at the right or left caudal vein. Three days following the first treatment, the second dose was given. Four mice were in each treatment group. Over the course of 26 days, measurements of the tumour’s length and width were taken using the Vernier calliper at regular intervals. The tumour volumes of each group of mice were shown as mean SEM in the data. The following formula was used to calculate the tumour volume (mm^3^):Tumour Volume (mm^3^) = ½ (length × width^2^).

Throughout the study, the gross body weight was also calculated and closely tracked. The mice were humanely killed by cervical dislocation after 26 days of treatment.

### 2.19. Blood Chemistry and Histology (H and E) Study

The blood was collected from the mice treated with PEG-SSNs-EGFR and without any treatment (control) on day 26 for biochemical analysis to calculate the levels of alkaline phosphatase (ALP), aspirate aminotransferase (AST), alanine aminotransferase (ALT), amylase (Amy), and creatinine (Cr) in serum samples of the mice. For histopathology analysis, collected major organs including heart, liver, spleen, kidney, and lung from both control and treatment groups were fixed at day 26. The fixed organs were then processed for hematoxylin and eosin (H and E) staining and images were taken for analysis.

### 2.20. Statistical Analysis

Statistical analysis was done using GraphPad software (San Diego, CA, USA) to calculate *p*-values. A *t*-test was applied for analyzing and comparing the significant difference in data obtained from in vitro and in vivo experiments. Data were considered statistically significant when * *p* < 0.05 and very significant when ** *p* < 0.001.

## 3. Results

### 3.1. Turbidity Analysis and Optical Imaging of PEG-SSNs

The turbidity of PEG-SSNs formulated by adding different concentrations of biotin-PEG (1 µL, 5 µL, 10 µL and 15 µL of 1 µM) to the preformed SSNs and incubating for 10 min at 37 **°**C (illustrated in Figure 1A) was measured through absorbance at 320 nm to confirm the formation of the particles. As shown in Figure 1B SSNs without biotin-PEG showed higher turbidity and the addition of the PEG derivative gradually reduced the turbidity at lower concentrations (20–200 nM) and increased to the level observed for SSNs at a higher concentration (300 nM). The lower turbidity was probably due to the interactions of biotin-PEG with the surface of SSNs, slowing down the rate of particle formation, whereas the subsequent increase in absorbance at a relatively higher PEG concentration could be owing to the combined effect of a decrease in particle formation and inhibition of particle aggregation. To observe the aggregation pattern of NPs, optical images of SSNs with or without PEG were taken (Figure 1C). PEGylated SSNs demonstrated the uniform distribution of the particles, particularly for 300 nM biotin-PEG used for the surface coating.

### 3.2. Characterization of PEG-SSNs by Dynamic Light Scattering (DLS)

Particle size and net surface charge are the two vital factors for determining the fate of NPs both in vitro and in vivo. The rate of biodistribution, tumour accumulation, and clearance of NPs may widely vary with particle size and surface charge [59,60,61]. As analyzed by the DLS method, the size of the SSNs prepared by mixing 40 mM of SrCl_2_ and 10 mM of Na_2_SO_3_ without PEG showed a mean particle size of 2096.33 nm and a zeta potential of −11.95 mV (Figure 2A,B). On the other hand, PEGylated SSNs’ average diameter showed a significant decreasing pattern at different biotin-PEG concentrations. The sizes of PEG-SSNs synthesized at varying concentrations of biotin-PEG (20–300 nM) ranged from 689 to 832 nm while the zeta potential varied from −9.345 to −7.25 mV, indicating the successful adsorption of the PEG moiety on the surface of SSNs.

On the other hand, size distribution based on intensity, number, and volume was evaluated by DLS technique along with poly dispersity index (PDI) [62]. As shown in Figure 2C: i, ii, and iii, SSNs revealed large particles based on intensity, number, and volume distribution with a PDI value of 0.319. In contrast, modification of SSNs by using 20 nM of biotin-PEG resulted in overall smaller particles with a PDI value of 0.869 in Figure 2C: iv, v, and vi.

### 3.3. Field Emission Scanning Electron Microscopy (FESEM) of PEG-SSNs

The morphology of SSNs with or without biotin-PEG was visualized by using FE-SEM. The shape of SSNs was almost spherical with a spiky surface (Figure 3a–c), and that of PEG-SSNs prepared with 20 nM of biotin-PEG (Figure 3d–f) remained the same with a reduction of a spiky surface due to entrapment of PEG chain on the surface.

The size of SSNs was found to be within the range of 1.28–1.67 µm, whereas that of PEG-SSNs was reduced significantly to 384–516 nm, indicating that PEG has a vital role in reducing the particle diameter by adsorbing onto the particle surface and preventing particle-particle aggregation. The smaller particles with reduced spiky nature of the surface of PEG-SSNs might play beneficial roles in inhibiting opsonization and macrophage uptake and thus improving the half-life of the formulation in the systemic circulation.

### 3.4. Elemental Analysis of PEG-SSNs by EDX

To analyze the elemental composition of SSNs and PEGylated SSNs formulated with 20 nM of biotin-PEG, both were subjected to EDX analysis. As shown in Figure 4, although both SSNs and PEG-SSNs contained C, O, Sr, and S elements, the latter contained a relatively higher percentage of C and O, accounting for the existence of biotin-PEG. Si and Pt were also detected along with C owing to the usage of carbon tape and glass holder and sample processing through Pt sputtering, prior to the observation and analysis.

### 3.5. Assessment of Plasma Stability and siRNA Binding Affinity of PEG-SSNs

The stability of PEG-SSNs in blood was determined by measuring particle size distributions in medium, water, and mice plasma. As shown in Figure 5A PEGylated SSNs in DMEM and deionized water displayed the same particle size distribution pattern with a PDI value of 0.659. On the other hand, in mice blood, the particles were found in a much smaller size, which might be due to the coating of blood proteins on the surface of PEG-SSNs. Moreover, the affinity of negatively charged siRNA towards the NPs is one of the major determinants for the successful delivery of the siRNA into the target cancer cells, preventing its early release in extracellular fluid including the blood, and consequential premature degradation by nucleases. Both SSNs and PEGylated SSNs were complexed with 10 nm of AF-488 fluorescence siRNA and centrifuged to sediment siRNA-loaded NPs and estimate the unbound, free siRNA in the supernatant. As shown in Figure 5B both SSNs and biotin-PEG-coated SSNs showed siRNA binding efficiency of more than 90%, indicat ing that PEG coating didn’t interfere with the siRNA binding.

### 3.6. Uptake of PEG-SSNs/AF-488 siRNA Nanocomplex into Human Breast Cancer Cells

Cellular internalization study of SSNs and PEG-SSNs with loaded AF-488 siRNA was performed into a human breast cancer cell line, MCF-7 using a confocal scanning laser microscope (CSLM) to observe the internalized siRNA-loaded NPs rather than the particles adsorbed onto the cell surface. Confocal images of the cells treated with only DMEM (control), free siRNA (control), and siRNA-loaded SSNs with or without PEG were analyzed after 1 h of the treatment. As shown in Figure 6, the intracellular green fluorescence intensity of AF-488 siRNA delivered with PEG-SSNs was significantly higher than the intensity of the free siRNA and the unmodified SSNs, confirming that the cellular uptake of small-sized PEGylated SSNs was more efficient than the unmodified SSNs.

### 3.7. Release Profile of SSNs and PEG-SSNs Complexes

The performance of nanoformulations to carry siRNAs relies on the efficient release of siRNA in the target cell cytoplasm to silence the target gene expression after cellular internalization. It is assumed that quick releases of siRNAs from endocytosed nanocomplexes at an early endosomal stage would exert rapid pharmacological effect and bypass the lysosomal degradation. Figure 7 represents the SSNs and PEG-SSNs/siRNAs complex’s release profile at different pHs starting from physiological pH to acidic pH. The release of fluorescence siRNA from SSNs and PEG-SSNs was increased by more than 60% as the pH dropped from 7.5 to 5.0. At pH 5.0 the PEG-SSNs showed a higher release of siRNAs than unmodified SSNs. This was probably due to the instability or dissociation of PEG-SSNs at acidic pH, releasing more siRNAs.

### 3.8. Biocompatibility Assay

The non-toxic nature of NPs is another vital factor for the successful delivery of siRNA in a clinical setting. A standard MTT colorimetric assay was carried out to evaluate the potential cytotoxicity of SSNs and PEG-SSNs in breast cancer cells MCF-7, 4T1, and healthy human embryonic kidney cells, HEK 293. Figure 8A demonstrated that SSNs with or without PEG showed no significant hampering of cell activity and that cell viability was maintained close to 80%, following exposure of the cells to the formulations for a consecutive period of 48 h, compared to the untreated cells. To further understand the biocompatibility of PEG-SSNs, a hemolytic assay was performed by measuring the level of hemoglobin released due to the rupture of the erythrocytes (RBCs) after treatment with the NPs. As shown in Figure 8B, no noticeable hemolysis was found for SSNs and PEG-SSNs compared to the control group.

### 3.9. In Vitro Antitumour Activity Assessment of EGFR siRNA-Coupled PEG-SSNs

EGFR and its activated signaling cascades have notable roles in developing and progressing human breast cancers. Overexpression of EGFR was reported in breast cancer including TNBC (triple-negative breast cancer) and other types of breast cancer [55,58,63,64,65,66,67]. We hypothesized that the silencing of EGFR expression using a specific siRNA might reduce [68] the proliferation and growth of breast cancer cells and be beneficial for exerting antitumour activity. To investigate the in vitro anticancer efficacy, SSNs and PEG-SSNs coupled with EGFR siRNA were incubated separately with MCF-7 and 4T1 cells with a view toward silencing the overexpressed EGFR; the cell viability was measured by MTT assay. It was found that treatment of MCF-7 and 4T1 with SSNs with loaded anti-EGFR siRNA for 48 h resulted in enhanced cytotoxicity of 17% and 4%, respectively (Figure 9). PEG-functionalized SSNs also showed a higher growth inhibitory effect in both cell lines than the empty NPs. These data indicate that both SSNs and PEG-SSNs efficiently carried the siRNA into the breast tumour cells and inhibited tumour cell growth by silencing EGFR gene expression.

### 3.10. EGFR Expression Study by RT-PCR

The silencing efficacy of SSNs with or without PEG after being complexed with anti-EGFR siRNA was evaluated by RT- PCR in MCF-7 cell lines. The result (Figure 10) clearly demonstrates that the expression of EGFR was lowered significantly in PEG-SSNs/EGFR-treated group compared to other tested formulations. Free SSNs didn’t reduce the level of EGFR while PEG-SSNs showed a bit of reduction of EGFR level. We assume that the quick sedimentation of PEG-SSNs on the cells affected the viability of cells, resulting in being slightly cytotoxic found in the biocompatibility test.

### 3.11. Protein Corona Profiling of PEG-SSNs by LC-MS/MS

Identification of the proteins adsorbed on the surface of SSNs and PEG-SSNs was made with the help of LC-MS/MS following the digestion of the collected proteins with trypsin. The peptides obtained from de novo sequencing were recognized as exact or homologous peptides using the Mus_musculus database (SwissProt). The protein corona profile was characterized by unique peptides, molecular weight, coverage % for peptides, and significance (−10 lgp). Detected proteins were listed along with their functions in Appendix A (the contents of the table were reported in our previous research article) [65] and Appendix A for SSNs and PEG-SSNs, respectively, and classified in a pie chart based on their biological functions. As shown in Figure 11A, SSNs prepared without biotin-PEG showed a predominant affinity for several types of proteins, including structural proteins (different keratins, Nup205), transport proteins (albumin, oligomeric Golgi complex subunit 7), and enzymes (protein kinases, endonucleases, glutamine synthetase) [58]. On the other hand, PEGylated particles were found to selectively bind structural proteins (keratins) and transport proteins (albumin). Serum albumin, the most abundant protein in the blood, and a dysopsonin, were detected on both SSNs and PEG-SSNs. Binding to a dysopsonin would confer the stealth properties to SSNs and PEG-SSNs and enable them to escape from the interactions with opsonins, thus reducing phagocytic uptake. Moreover, the hydrophilic coating of SSNs with PEG was found to significantly reduce the diversity of the proteins attached to the PEG-SSNs compared to the unmodified SSNs.

### 3.12. Bio-Distribution of PEG-SSNs

AF 488 siRNA was coupled with SSNs and PEG-SSNs and the resultant complexes were subjected to a biodistribution study following intravenous administration to breast tumour-bearing mice. After 4 and 24 h of the injection, major organs and tumours were harvested to quantitate the fluorescence intensity in the homogenized tissues. Two concentrations of biotin-PEG (20 and 200 nM) were used to fabricate the PEGylated SSNs in order to see the effects of different degrees of PEGylation on the biodistribution. SSNs without any PEG association showed significant accumulation in the brain and spleen with minimal tumour accumulation. At 4 h, fluorescence signals were found to be increased in the tumour along with the brain and spleen (Figure 11B). On the other hand, coating SSNs with 20 nM of biotin-PEG resulted in higher accumulation in the tumour than free SSNs after 4 h of treatment. Moreover, higher uptake by kidneys and brain were also observed at this time, indicating that smaller sizes of PEGylated SSNs might assist in transporting them to the blood vessels within the brain as well as the kidneys for excretion [69,70]. Interestingly at 24 h, remarkably high tumour accumulation was noticed for PEG-SSNs made with 20 nM of biotin-PEG, as shown in Figure 11B. The siRNA from the brain was eliminated after 24 h, which might rule out any potential neurotoxicity. At a higher concentration of PEG (200 nM), no fluorescence signals were detected in any major organs or the tumour, which might be due to the clearance of the highly PEGylated particles from systemic circulation owing to their small size and high hydrophilicity.

### 3.13. Tumour Regression Study of EGFR siRNA/PEG-SSNs Complexes in a Mouse Model

To investigate the anti-tumour effects of EGFR siRNA-loaded PEGylated NPs, tumour bearing mice were treated with SSNs, PEG-SSNs, EGFR siRNA/SSNs, and EGFR siRNA/PEG-SSNs through the tail vein. Tumour volumes were regularly measured at a particular time interval till the 26th day of 4T1 cells inoculation into the mammary pad. As shown in the tumour growth curve (Figure 12), the untreated group’s tumour volume was ~1956.15 mm^3^ on the last day and the growth rate was faster than other groups of mice. ‘SSNs’ and ‘PEG-SSNs’ groups also showed large tumour volumes (~1606–1756 mm^3^) with an increasingly growing trend throughout the period. SSNs complexed with EGFR siRNA demonstrated a modest tumour growth compared to the other three groups, with an average tumour volume of ~1182.80 mm^3^.

However, PEGylated SSNs with complexed anti-EGFR siRNA showed much slower tumour growth rate, exerting significantly high tumour inhibitory effects with the average tumour volume of ~726.46 mm^3^ (almost 60% reduction), compared to the control and empty NPs treatment groups. The tumours were excised after 26 days of tumour inoculation, and it was found that the average tumour weight of the group treated with EGFR/PEG-SSNs complexes had the most negligible tumour weight compared to the control and empty NPs groups.

### 3.14. Safety Evaluation of PEG-SSNs

The primary safety marker, i.e., the body weight of mice was the same for all groups, both treated and untreated throughout the experiment (Figure 13A). Additionally, the blood and major organs were collected from both ‘control’ and ‘EGFR siRNA/PEG-SSNs’ groups for biochemical and histological analysis after 26 days of tumour inoculation. As shown in Figure 13B, the level of ALP, creatinine, and ALT were found to be normal for both control and treatment groups except for a slight increase in AST level. On the other hand, no significant changes were observed in the histology of major organs of the mice treated with EGFR siRNA/PEG-SSNs (Figure 13C).

## 4. Discussion

The advent of inorganic nanoparticles for transporting a specific siRNA opens a new window for silencing an overexpressed target oncogene as an innovative therapeutic approach for cancer management. Although inorganic NPs are advantageous in terms of drug loading capability, unique physicochemical properties, and biocompatibility, their instability in the systemic circulation and off-target bio-distribution remain a major challenge [71,72,73]. The main goal of this study was to investigate the feasibility of PEGylated SSNs in delivering siRNA into target breast cancer cells and their therapeutic potential in a breast tumour mouse model. Previously we reported SSNs as pH-sensitive inorganic nanocarriers for the delivery of siRNA into breast cancer cells both in vitro and in vivo [58]. However, SSNs showed non-specific distribution and low tumour accumulation of the intravenously administered siRNA. In the present study, we have modified the surface of SSNs with biotin-PEG with a view to prolonging circulation half-life, reducing off-target distribution, and increasing tumour accumulation of siRNA.

The physicochemical properties of NPs like particle size distribution and surface charge affect the interactions between blood proteins and the NPs, the recognition of opsonin-coated NPs by RES, their extravasation from blood vessels to tumour tissues, and clearance by kidneys after systemic administration [6,60,74,75,76,77]. PEGylated SSNs were successfully fabricated with an average particle dimension much smaller than that of SSNs (Figure 2). The surface charge of SSNs particles decreased to more electropositive in the presence of biotin-PEG, indicating PEG adsorption on the surface of SSNs (Figure 2). The overall small-sized and hydrophilic PEG-SSNs were assumed to prevent their interactions with differently charged blood proteins, minimizing the RES clearance, improving the colloidal stability in the systemic circulation, and thus increasing the delivery of siRNA into the target tumour cells [78,79]. The morphology and size analysis by FE-SEM identified PEG-SNNs as smaller particles with reduced spiky nature compared to SSNs (Figure 3), which could be due to the effect of PEG in reducing particle-particle interactions via ionic interactions. A previous study from our laboratory revealed that carbonate apatite (CA) NPs coated with biotin-PEG reduced the particle size significantly [70]. Even though SSNs are quite different from CA NPs with respect to size, morphology, and surface charge, the effect of PEGylation on notably reducing the diameter of both particle types is quite interesting. The biotin moiety in PEG contains protonable amine groups, which seemingly allowed PEG to interact with cationic domains (Sr++) of SSNs. A, incorporating biotin-PEG reduced the spiky surface feature of SSNs, which might prevent the non-specific protein entrapment to the NPs [80,81]. We, therefore, hypothesized that PEGylated SSNs would confer better pharmacokinetic and pharmacodynamic advantages owing to their smaller particle size with unique surface properties, compared to the unmodified SSNs.

Next, we examined the elemental composition of generated SSNs and PEG-SSNs through EDX analysis. Both surface-modified and unmodified SSNs were found to possess Sr, S, and O, reflecting the successful formation of SSNs (Figure 4). Additionally, C and O were found at a relatively higher percentage in PEG-coated SSNs than SSNs, indicating an association of biotin-PEG with the former (Figure 4). Since maintaining the stability of PEG-SSNs in blood is critical, we measured the particle size distribution and PDI value of PEG-SSNs in water, media, and mouse plasma. The size and PDI value remained the same in water and media. Although the PDI value was also unchanged in blood plasma, smaller-sized particle distribution was found. This could be due to the spontaneous coating of PEG-SSNs with the mouse plasma proteins (Figure 5A).

The siRNA binding efficiency and stability of siRNA-nanocomplexes are two key factors for efficient siRNA transport to the target cells. The NPs can protect enzymatic degradation of siRNA by electrostatically binding it, thereby facilitating cellular internalization of the siRNA via endocytosis. We observed that significant siRNA binding efficiency of SSNs and PEGylation did not hamper the binding (Figure 5B). In line with the significant siRNA binding affinity of PEG-SSNs as well as their smaller sizes, we found that PEG-SSNs mediated an augmented cellular entry of fluorescent siRNAs compared to SSNs and free siRNA, as demonstrated by strong fluorescence intensity after 1 h of cell treatment (Figure 6).

We hypothesized that after being internalized into the cell through endocytosis, pH-sensitive PEG-SSNs like SSNs [45] and carbonate apatite NPs [82,83,84,85,86] would be dissolved and release the delivery cargo at early endosomal stages. The release pattern of fluorescence siRNA at different pHs starting from physiological pH 7.5 to acidic pH 5.0 revealed the release of the highest amount of fluorescence siRNA at acidic pH (Figure 7). We hypothesized that SSNs or PEG-SSNs would completely dissolve at acidic pH; releasing cations and anions in endosomes. The consequentially elevated osmotic pressure across the endosomal membrane resulted in swelling and rupture of the membrane and release of the siRNA in the cytosol to effectively cleave a target mRNA. Both hemolytic activity and cytotoxicity studies of SSNs and PEG-SSNs showed no significant toxic effects in blood cells as well as in healthy cells and breast cancer cell lines (Figure 8), indicating PEG-SSNs are non-toxic and biocompatible transporters.

Since EGFR is critically involved in activating the intracellular downstream signaling cascades, such as PI-3 kinase and MAPK signaling pathways, leading to proliferation and survival of breast cancer cells [66,87,88,89,90], treatment of MCF-7 and 4T1 cells with EGFR targeting anti-siRNA-loaded PEG-SSNs and SSNs for a consecutive period of 48 h was found to significantly kill the cancer cells (Figure 9). The augmented anti-tumour effects could be due to the efficient cellular uptake of the siRNA/NPs complexes and the early release of the siRNA from the endosome, thereby silencing the expression of EGFR genes and inducing the apoptosis in the treated cells [91,92]. To reconfirm the silencing efficacy of NPs/EGFR siRNA complexes in MCF-7 cells, the ratio of EGFR gene expression was checked using RT-PCR (Figure 10). The data showed a significant decrease in the level of EGFR expression in the cells treated with PEG-SSNs. The lower expression of EGFR apparently led to killing the cancer cells, thereby decreasing the viability of MCF-7 cells as reflected in the MTT assay.

Once NPs enter the blood circulation after intravenous administration, they are exposed to thousands of blood proteins and other biomolecules owing to their high surface energy [93]. In particular, the plasma proteins bind to the exogenous NPs, forming ‘protein corona’ on their surface, which immensely modulates biodistribution, therapeutic effects, and the toxicity of NPs [94,95,96]. When the protein corona contains opsonin as one of its constituents, the NPs undergo opsonin-triggered phagocytosis and accumulate in RES organs like the liver, spleen, and lungs, resulting in off-target distribution and rapid clearance from the circulation. Some vital factors determine the amount and diversity of the proteins present in the corona, such as the physicochemical properties of NPs, exposure time, and abundance of plasma proteins. It was found that NPs with hydrophobic and highly positive or negative surface charge demonstrated a higher tendency to form a protein corona than hydrophilic and neutral-charged particles [25,97]. NPs coated with PEG could protect the NPs from interacting with the plasma proteins through stealth effects which are highly dependent on PEG length and density [98,99]. As shown in Appendix A, unmodified SSNs showed affinity towards different kinds of proteins, leading to the distribution of the particles to RES organs and other vital organs. This might lower the amount of siRNA-nanocomplex in the circulation (i.e., shortening its half-life) owing to the rapid clearance, thus reducing the transportation of an adequate amount of siRNA into the target tumour site. The cationic domain of SSNs (Sr^2+^) could interact electrostatically with negatively charged proteins available in the blood plasma, reducing their stability in blood. On the other hand, surface-functionalization of SSNs with PEG resulted in less entrapment of the proteins entrapment on their surface, as shown in Appendix A and Figure 11A. The hydrophilic PEG layer on SSNs seemingly hindered the adsorption of different types of proteins on the particle surface due to the reduction of surface charge. Less interaction with opsonin proteins leads to higher stability in plasma, which might improve the accumulation in the target tumour cells. Most importantly, PEG-SSNs showed affinity towards albumin, a dysopsonin might skip the PEG-SSNs from macrophage recognition, prolonging the circulation time. Although both SSNs and PEG-SSNs have an affinity toward the dysopsonin, the latter showed less binding affinity towards other proteins than the former. Thus, PEGylated SSNs were expected to be accumulated more in tumours and less in other organs, compared to SSNs.

Then we checked the organ distribution of fluorescent siRNA-loaded SSNs and PEG-SSNs in mice to find any correlation between the composition of protein corona formed on SSNs and PEG-SSNs and their biodistribution profiles. Figure 11B shows that unmodified SSNs were found to be accumulated in the brain and RES organ, i.e., the spleen, apparently triggered by some constituent proteins in the corona. In contrast, PEG-SSNs reduced the siRNA accumulation in the RES organs and brain while tumour targetability was enhanced significantly after 24 h of treatment compared to unmodified SSNs. We assumed that by reducing opsonization and promoting dysopsonin binding, PEG-SSNs could escape the opsonin-mediated macrophage uptake, inhibiting the rapid clearance and off-target distribution of the siRNA. Another important factor that affects the biodistribution of siRNA is the size of the particles. The pore cutoff sizes of tumour vasculature ranging from 200 nm to 1.2 µm [100] allow the small-sized siRNA/PEG-SSNs complex to penetrate blood vessels of the tumour region and be retained into the tumour tissue via EPR effect.

Finally, to test the therapeutic potential of PEGylated SSNs as a tool for functional siRNA delivery in vivo, both SSNs and PEG-SSNs were electrostatically associated with EGFR siRNA and tested in 4T1 tumour-bearing mice for evaluating anticancer effects in a pre-clinical setting. PEG-SSN-EGFR complex showed a significant reduction in tumour volume compared to the control groups, which could be attributed to the extended biological stability of NPs, heightened tumour accumulation, improved cellular internalization, and fast release of the siRNA inside the cancer cells to silence the overexpressed EGFR. Free siRNA which is prone to systemic degradation by nucleases and subjected to rapid clearance by kidneys showed no notable tumour inhibitory effects [101,102,103,104]. With reduced particle size and a protein corona rich in a dysopsonin (albumin), and devoid of many other proteins (detected on SSNs), PEGylated SSNs might have prevented not only siRNA degradation by RNases, but also RES clearance during traveling in systemic circulation. In addition, the small-sized PEG-SSNs mediated more tumour accumulation by EPR effect and cellular internalization of siRNA into target cancer cells, resulting in a remarkable inhibition of breast tumour growth via the silencing of EGFR expression in the tumour cells (Figure 12A). The unchanged body weight of the mice treated with free SSNs, and siRNA-coupled SSNs indicates no apparent systemic toxicity of NPs and NP-siRNA complexes in the mouse model. There were no notable changes in the blood biochemical parameters in control or treated mice. Additionally, no noticeable damages or changes in the histology of major organs collected from both control and treatment groups were observable, indicating a high safety profile of PEG-SSNs. These results signify the feasibility of PEG-SSNs as a promising siRNA transporter to suppress tumourigenesis in vivo by enabling efficient knockdown of the target mRNA(s).

## 5. Conclusions

Novel PEG-grafted spherical inorganic SSNs of around 400 nm were synthesized with excellent biocompatibility and serum stability. The small-sized and less aggregated SSNs demonstrated high siRNA encapsulation efficiency and augmented cellular uptake in breast cancer cells. Immobilization of biotin-PEG onto SSNs led to negligible protein adsorption on SSNs, reduced off-target distribution apparently by preventing clearance by RES, reduced toxicity, and finally, augmented tumour accumulation of siRNA in a breast cancer mouse model. In terms of therapeutic potency, PEGylated SSNs carrying electrostatically associated anti-EGFR siRNA significantly resisted the growth of the tumour and reduced its mass without any significant systemic toxicity in the animal model, reducing the dose of the siRNA applied and ultimately lowering the cost of treatment. Taken together, outperformed PEG-SSNs can be further tuned according to patient demands and may be considered a leading candidate for human cancer management in the near future.

## Figures and Tables

**Figure 1 jfb-13-00211-f001:**
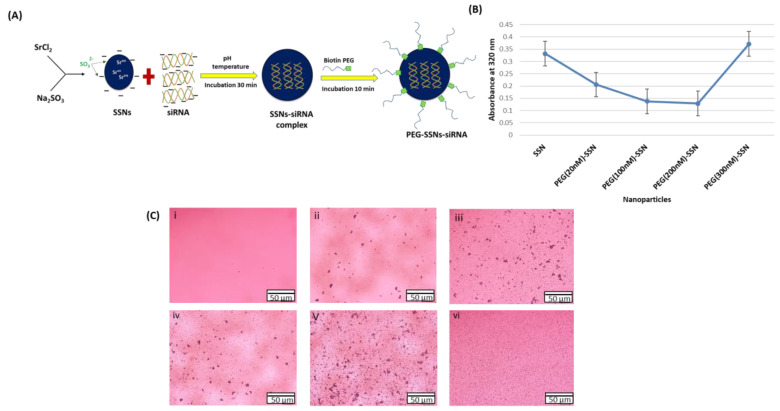
Fabrication and characterization of PEG-SSNs through turbidity analysis and optical microscopy. (**A**) A schematic representation of the preparation of PEG-coated SSNs/siRNA complexes; (**B**) Turbidity of SSNs with or without biotin-PEG at different concentrations (20 nM, 100 nM, 200 nM, and 300 nM) measured through absorbance reading at 320 nm; (**C**) Optical images of SSNs with or without PEG-media only (i), SSNs formulated with 40 mM of SrCl_2_ and 10 mM of Na_2_SO_3_ (ii), PEG-SSNs prepared with 40 mM of SrCl_2_ and 10 mM of Na_2_SO_3_ and coated with 20 nM (iii), 100 nM (iv), 200 nM (v), and 300 nM (vi) of biotin-PEG.

**Figure 2 jfb-13-00211-f002:**
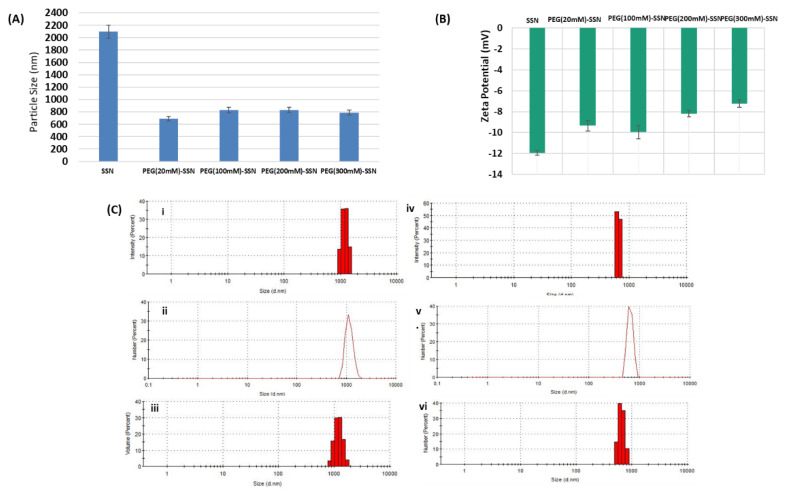
Sizes, zeta potential, and particle size distributions of SSNs with or without PEG. (**A**) Sizes and (**B**) zeta potential of SSNs and PEGylated SSNs prepared in presence of 20, 100, 200, and 300 nM of biotin-PEG. (**C**) Particle size distributions by intensity, number, and volume of SSNs (i–iii) and PEG-SSNs (iv–vi) prepared by coating SSNs with 20 nM of biotin-PEG. SSNs were prepared with 40 mM of SrCl_2_ and 10 mM of Na_2_SO_3_.

**Figure 3 jfb-13-00211-f003:**
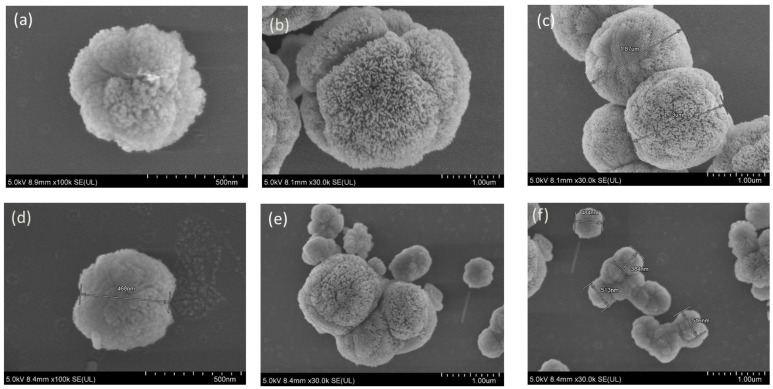
FE-SEM images of SSNs and PEG-SSNs. SSNs were prepared by mixing 40 mM of SrCl_2_ and 10 mM of Na_2_SO_3_ in 50 µL of aqueous solution and PEG-SSNs were synthesized by adding 20 nM of biotin-PEG to SSNs. Images denoted by (**a**–**c**) represent unmodified SSNs, while those indicated by (**d**–**f**) represent PEGylated SSNs. All images were observed at 5.0 KV.

**Figure 4 jfb-13-00211-f004:**
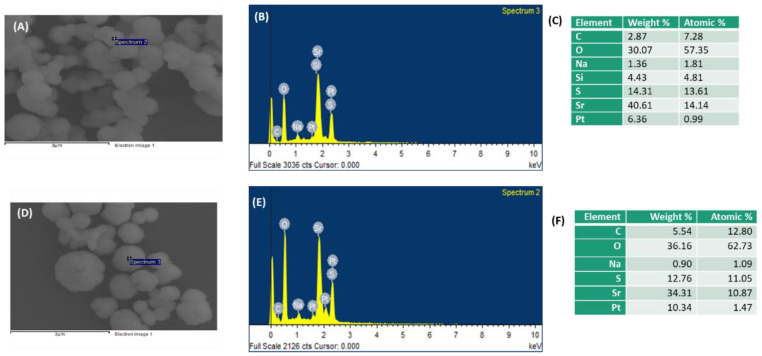
Elemental analysis of SSNs and PEG-SSNs by EDX. SSNs were prepared by mixing 40 mM of SrCl_2_ and 10 mM of Na_2_SO_3_ in 50 µL of an aqueous solution, while PEG-SSNs were synthesized by adding 20 nM biotin-PEG to the SSNs. (**A**,**B**) EDX spectrum of SSNs; (**C**) weight and atomic percentage of SSNs; (**D**,**E**) EDX spectrum of PEG-SSNs and (**F**) weight and atomic percentage of PEG-SSNs.

**Figure 5 jfb-13-00211-f005:**
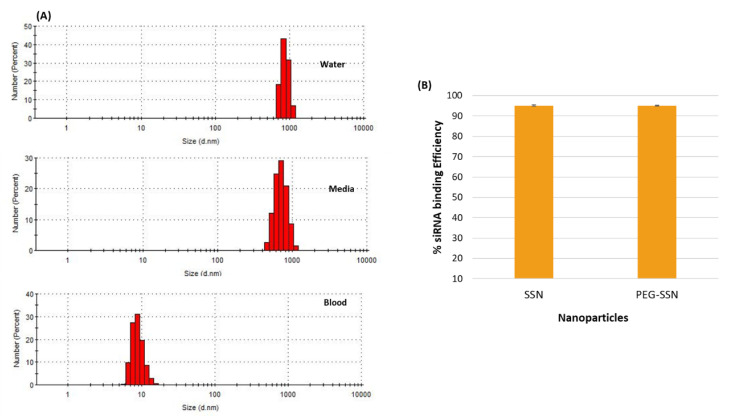
(**A**) Particle size distribution analysis for PEG-SSNs in water, DMEM, and the blood obtained from mice by dynamic light scattering (DLS). SSNs were prepared by mixing 40 mM of SrCl_2_ and 10 mM of Na_2_SO_3_ in 50 µL of an aqueous solution, while PEG-SSNs were synthesized by adding 20 nM biotin-PEG to the SSNs, prior to the addition to water, DMEM, and mouse plasma. (**B**) Estimation of siRNA binding efficiency of SSNs and PEG-SSNs after complexation with AF-488 siRNA. SSNs and PEG-siRNA-SSNs were prepared in presence of 10 nM of AF-488 fluorescence siRNA in 50 µL of an aqueous solution. After the samples were centrifuged at 13,000× *g* rpm for 15 min at 4 °C, the supernatants were collected and fluorescence intensity was measured.

**Figure 6 jfb-13-00211-f006:**
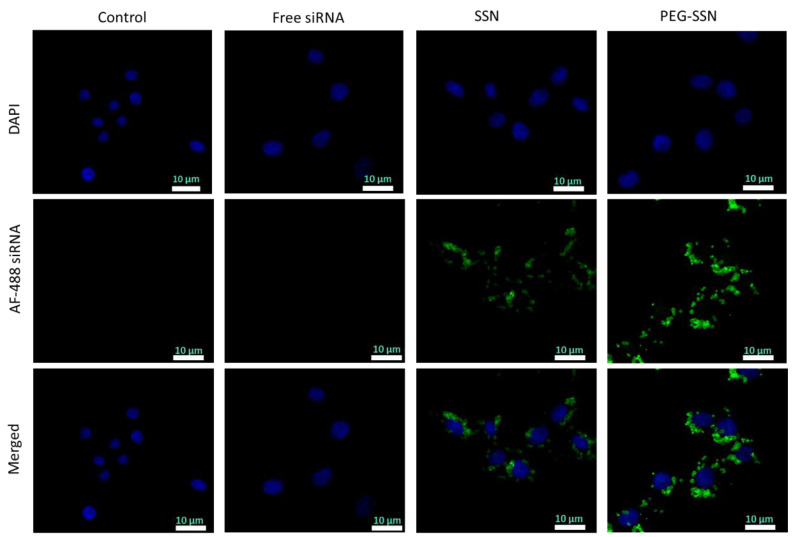
Cellular uptake of free AF-488 fluorescent siRNA, SSNs, and PEG-SSNs with loaded fluorescent siRNA in MCF-7 cells. SSNs were prepared by mixing 40 mM of SrCl_2_ and 10 mM of Na_2_SO_3_ in 50 µL of an aqueous solution while PEG-SSNs were synthesized by adding 20 nM biotin-PEG to the pre-formed SSNs. CSLM images were taken after 1 h incubation of the treated cells. AF-488 fluorescence siRNA was represented by green color while the nuclear staining DAPI by blue color.

**Figure 7 jfb-13-00211-f007:**
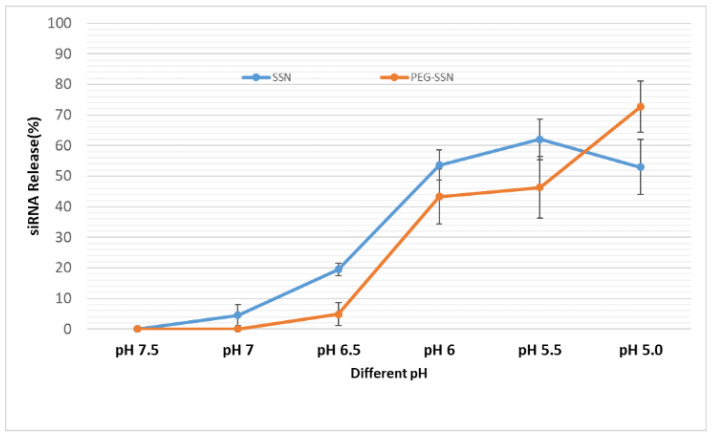
The release profile of AF-488 fluorescence siRNA from SSNs and PEG-SSNs at different pH. SSNs were prepared by mixing 40 mM of SrCl_2_ and 10 mM of Na_2_SO_3_ in 50 µL of aqueous solution while PEG-SSNs were synthesized by adding 20 nM biotin-PEG to the pre-formed SSNs.

**Figure 8 jfb-13-00211-f008:**
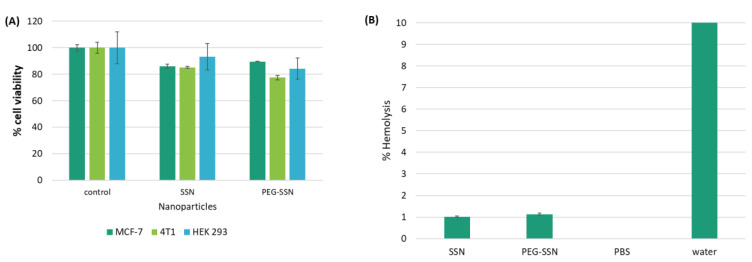
(**A**) Cell viability of SSNs and PEG-SSNs in MCF-7, 4T1, and HEK 293 cells and (**B**) Hemolysis percentage of RBCs incubated with SSNs with or without PEG. SSNs were prepared by mixing 40 mM of SrCl_2_ and 10 mM of Na_2_SO_3_ in 50 µL of an aqueous solution, while PEG-SSNs were synthesized by adding 20 nM biotin-PEG to the SSNs. Data were represented as mean ± SD for triplicate samples.

**Figure 9 jfb-13-00211-f009:**
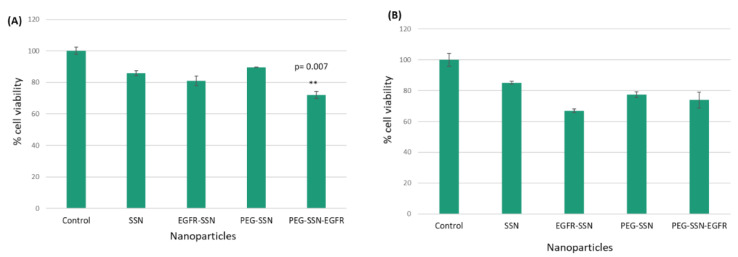
Cytotoxicity assay of SSNs and PEG-SSNs with loaded EGFR siRNA (denoted here as ‘EGFR’) in MCF-7 (**A**) and 4T1 (**B**) cells. SSNs were prepared by mixing 40 mM of SrCl_2_ and 10 mM of Na_2_SO_3_ in the absence or presence of 1 nM siRNA in 50 µL of an aqueous solution, while PEG-SSNs were synthesized by adding 20 nM biotin-PEG to the SSNs previously formed in absence or presence of the same dose of the siRNA. Cell viability was assessed after treatment of MCF-7 (**A**) and 4T1 cells (**B**) with SSNs, SSNs-EGFR, PEG-SSNs, and PEG-SSN-EGFR for a consecutive period of 48 h. Values were considered very significant (**) at *p*-value 0.001 to 0.01 and significant (*) at *p*-value 0.01 to 0.05 compared to SSNs and PEG-SSNs.

**Figure 10 jfb-13-00211-f010:**
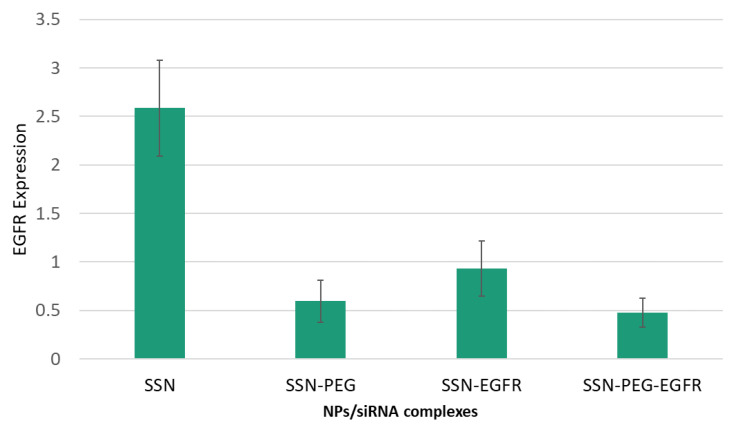
The relative expression of EGFR in MCF-7 cells treated with SSNs, PEG-SSNs, EGFR-SSN, and EGFR-PEG-SSNs determined by using RT- PCR. SSNs were prepared by mixing 40 mM of SrCl_2_ and 10 mM of Na_2_SO_3_ in the absence or presence of 1 nM of anti-EGFR siRNA in 50 µL of an aqueous solution, while PEG-SSNs were synthesized by adding 20 nM biotin-PEG to the SSNs.

**Figure 11 jfb-13-00211-f011:**
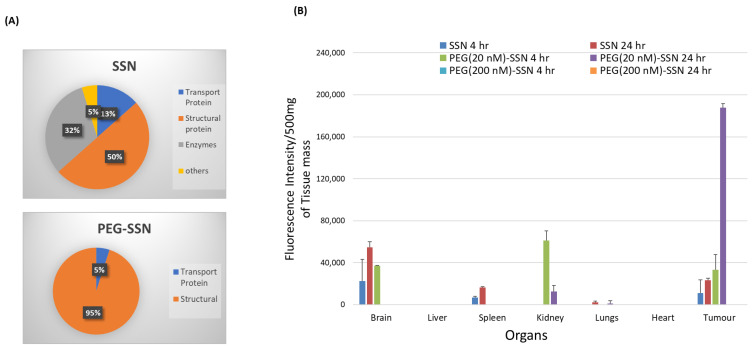
(**A**) Protein corona composition of SSNs and PEG-SSNs. SSNs were prepared by mixing 40 mM of SrCl_2_ and 10 mM of Na_2_SO_3_ in 50 µL of an aqueous solution, while PEG-SSNs were synthesized by adding 20 nM biotin-PEG to the SSNs, prior to incubation with 10% of mice plasma at 37 °C for 15 min. After centrifugation, the supernatant with unbound proteins was discarded and the pellet containing NPS with adsorbed proteins was washed and subjected to dissolution with EDTA to release the proteins for their subsequent digestion with trypsin and analysis by LC-MS. (**B**) Organ distribution of AF 488 siRNA complexed with SSNs and PEG-SSNs after 4 h and 24 h of intravenous injection into breast tumour-bearing female Balb/c mice. siRNA/SSNs were prepared by mixing 75 nM AF 488 siRNA, 40 mM of SrCl_2_, and 10 mM of Na_2_SO_3_ in 50 µL of an aqueous solution, while siRNA/PEG-SSNs were synthesized by adding 20 and 200 nM of biotin-PEG to siRNA/SSNs. Each group consisted of four mice and data were represented as mean ± SEM.

**Figure 12 jfb-13-00211-f012:**
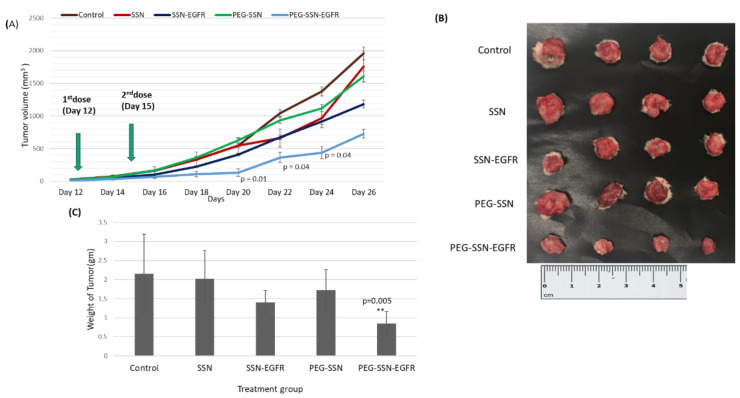
Tumour growth inhibitory effects of PEG-SSNs coupled with EGFR siRNA in a murine 4T1 subcutaneous tumour model (**A**). Tumour growth curves for five different groups treated with control (untreated), SSNs, EGFR siRNA/SSNs (denoted as ‘SSNs-EGFR’), PEG-SSNs, and EGFR siRNA/PEG-SSNs (denoted as ‘PEG-SSNs-EGFR’), (**B**) images of excised tumours at day 26, (**C**) weight of tumours collected at the end of the experiment. Each group of treatment contained four mice and error bars are based on the standard error of the mean. Values for PEG-SSNs-EGFR were very significant (**) at *p*-value of 0.001 to 0.01 and significant (*) at *p*-value of 0.01 to 0.05 compared to the PEG-SSNs group.

**Figure 13 jfb-13-00211-f013:**
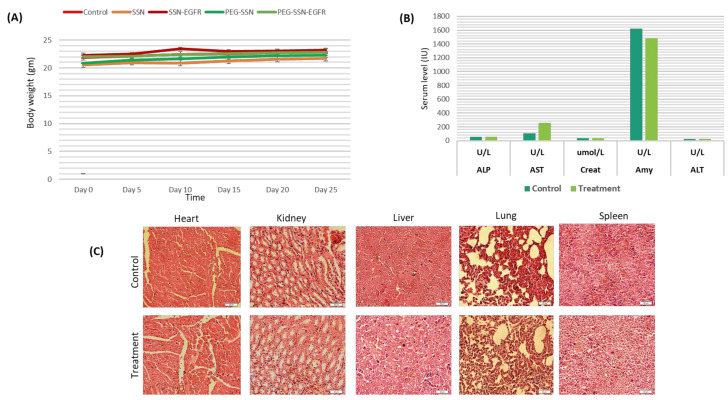
(**A**) Body weights of mice (**B**) Biochemical analysis of blood collected from control and treated mice after the 26th day of 4T1 cells inoculation into the mammary pad. (**C**) Histopathological images after H and E staining of major organs were collected from the control (untreated) and treatment groups on day 26. Scale bar, 20 µm.

## Data Availability

Not applicable.

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
