# Peer review of "PEGylated Strontium Sulfite Nanoparticles with Spontaneously Formed Surface-Embedded Protein Corona Restrict Off-Target Distribution and Accelerate Breast Tumour-Selective Delivery of siRNA"

_jfb, 2022, doi:10.3390/jfb13040211_

Round 1
Reviewer 1 Report
Check the grammar. There are minor mistakes as "for complete synesis of the latter"
The sentence "The biotechnology revolution has brought forward RNA interference (RNAi) as a potential therapeutic option over classical chemotherapy in cancer management." needs references
The sentence "Short inter-34 fering RNA (siRNA) is an efficient approach to silence expression of endogenous genes 35 responsible for cancer development." needs references
The paragraph "However, stable delivery of naked siRNA into bio-36 logical systems is quite challenging due to its short half-live in the in blood as a result of 37 nuclease-mediated degradation and renal clearance, poor cellular internalization rate 38 and potential off-target effects (1-5). Aiming to surmount these barriers and augment 39 siRNA's performance, a lot of synthetic transporters have been designed for safely carry-40 ing the siRNA into the cytoplasm of target cancer cells." It is necesarry to clarify that tumor targeting is completly different from cell targeting to avoid missleading information. Therefore, this paragraph must be changed. Moeover, the last sentence needs references.
The sentence "Both organic and inorganic nanoparticles (NPs) have been fabricated and investigated, especially in cancer gene therapy." needs references.
The paragraph "The last few decades have seen the rapid development of cancer nanotherapeutics, revealing the challenges to the safe and effective delivery of potential therapeutic NPs in cancer patients." needs references and must be rewritten. There are very few NPs in the clinical field, because of their limitations.
About the paragraph "Most of the intravenously administered NPs that accumulate in the tar-48 get tumor cells extravasate from the leaky tumor vasculature through enhanced permea-49 bility and retention (EPR) effects (8-11)." there is controversial information that should be mentioned. DOI: 10.1039/d1bm01398j
The paragraph "However, during traveling from the site of ad-50 ministration to the site of action, NPs encounter extracellular components of biological 51 systems, that determine the fate and navigating pattern of injected NPs inside the body." must mention the extraordinary work of PF. Joseph W. Nichols about this topic.
The sentence "Depending on the physicochemical properties and exposure time of NPs for exchanging 62 proteins, PC might be hard or soft." needs references
The sentence "Consequently, proteins-adsorbed NPs are subjected to the opsonin-mediated uptake by macrophage of reticuloendothelial system (RES), non-specific biodistribution, rapid clearance and formation of thrombosis, declining therapeutic efficiency of drug-loaded NPs." needs references
The paragraph "From the pharma-83 ceutical point of view, premature degradation before reaching the target site results in 84 the drug's sub-therapeutic action. So, it is necessary to minimize the unwanted clearance 85 of NPs for maximizing their therapeutic potential and targetability, while reducing the 86 off-target effects." needs references.
The paragraph "A surface modification would be a potential approach to avoid opsonization and sys-88 temic instability of NPs for improved therapeutic action and reduced side effects. Poly-89 ethylene glycol (PEG), a water-soluble, non-ionic and biocompatible coating material, is 90 widely used to extend the blood circulation time of drug-loaded NPs and thus enhance the therapeutic efficacy (30-34). PEGylation is the most acceptable way to reduce the 92 protein binding site on NPs, thus inhibiting the macrophage uptake via opsonin recep-93 tors and extending the half-life of the NPs in blood. Moreover, it also increases the size 94 and flexibility of NPs, reducing their renal filtration, and enhancing the extravasation by 95 forming softer NPs (35-38)." presents repeated information. Also the foloowing reference must be added. DOI: 10.1039/c7nr01607g
Section 2.7. Stability of PEG-SSNs in mice plasma: Can authors confirm Viscosity and refractive index than they used for these measuremnts? Moreover, authors are measuring diluted plasma, not real samples. Can authors check in non-diluted plasma?
Authors descrbe in section 2.8 siRNA encapsulation efficiency of PEG-SSNs. However, is a physical absorption, not an encapsulation. Please correct it.
There is lack of a releasing profile. Authors can check AF 488 fluorescence along the time in PBS or/and plasma.
If would be nice if authors perform more cytotoxicity test, in addition to hemolysis and MTT. There is lack of description of the positive control in the MTT., ethanol? Triton-X? Moreover, in order to evaluate the side effects of these new nanoparticles, the cytotoxicity over healthy cells is necessary.
Section 2.15. The authors do not extract all the blood from the animals before their sacrifice. They also homogenize the tissues before measuring fluorescence. How do you know that the fluorescence comes from particles that are truly accumulated and not from those present in blood? Even more important, why authors do not use AF488 as control? For in vivo experiments, is necesary 2 controls: empty NPS and drug alone.
In section 2.15, authors describe "When the tumor volume reached 75 mm3 on day 11 of injection", while in section 2.16, authors describe "After the tumor reached an average volume of ~22 mm3 at around day 10-11 of subcuta-311 neous injection". Why did they use a lower size? why did tumors grow slowly? Moreover, the utilization vernier calipers to estimate tumor volume, which lacks accuracy and can lead to overestimation of tumor reduction, that is, underestimation of total tumor volume. Tumor masses are usually infiltrative and often show ingrowth patterns that require the use of medical imaging techniques to accurately determine tumor volume.
Section 3.1.
TEM images in each step is necessary.
Since all DLS are above 800 nm. How do authors explain the application of these NPs in preclinical models of cancer, since 200 nm have been extensively described as the upper limit?
Figure 3. Please show all images with the same magnification.
Figure 5. TEM of NPs revealed more than 500 nm, but DLS in blood is around 10. In my experience it is posible only by 2 ways: 1. Digestion Ripenning. 2. A very low concentration of NPs, and thwe measurement of the proteins (with sizes around 10 nm). Please check by TEM and discuss
Section 3.6. TEM or ICP of cells is necessary to confirm that NPs have been uptaken, and not the siRNA is released, from NPs, outside of cells and is internalized alone.
Figures 7 and 8 exhibited almost the same information. Please combine them.
Since 99% of 99% of strontium in the human body is localized in bones and teeth, the ICP of the tissues help to better understand the biodistribution.
Figure 11c present a low magnification, and the tisssues can nor be properly observed. Please correct it.
Author Response
Reviewer #1:
The sentence "The biotechnology revolution has brought forward RNA interference (RNAi) as a potential therapeutic option over classical chemotherapy in cancer management." needs references
Responses: We have added reference.
The sentence "Short inter-34 fering RNA (siRNA) is an efficient approach to silence expression of endogenous genes 35 responsible for cancer development." needs references
Responses: We have added reference.
The paragraph "However, stable delivery of naked siRNA into bio-36 logical systems is quite challenging due to its short half-live in the in blood as a result of 37 nuclease-mediated degradation and renal clearance, poor cellular internalization rate 38 and potential off-target effects (1-5). Aiming to surmount these barriers and augment 39 siRNA's performance, a lot of synthetic transporters have been designed for safely carry-40 ing the siRNA into the cytoplasm of target cancer cells." It is necesarry to clarify that tumor targeting is completly different from cell targeting to avoid missleading information. Therefore, this paragraph must be changed. Moeover, the last sentence needs references.
Responses: We have modified the sentence and added the reference.
The sentence "Both organic and inorganic nanoparticles (NPs) have been fabricated and investigated, especially in cancer gene therapy." needs references.
Responses: We have added references.
The paragraph "The last few decades have seen the rapid development of cancer nanotherapeutics, revealing the challenges to the safe and effective delivery of potential therapeutic NPs in cancer patients." needs references and must be rewritten. There are very few NPs in the clinical field, because of their limitations.
Responses: we have rewriten the sentence and added reference.
About the paragraph "Most of the intravenously administered NPs that accumulate in the tar-48 get tumor cells extravasate from the leaky tumor vasculature through enhanced permea-49 bility and retention (EPR) effects (8-11)." there is controversial information that should be mentioned. DOI: 10.1039/d1bm01398j
Responses: We have modified the sentence.
The paragraph "However, during traveling from the site of ad-50 ministration to the site of action, NPs encounter extracellular components of biological 51 systems, that determine the fate and navigating pattern of injected NPs inside the body." must mention the extraordinary work of PF. Joseph W. Nichols about this topic.
Responses: We have mentioned the work.
The sentence "Depending on the physicochemical properties and exposure time of NPs for exchanging 62 proteins, PC might be hard or soft." needs references
Responses: We have added the reference.
The sentence "Consequently, proteins-adsorbed NPs are subjected to the opsonin-mediated uptake by macrophage of reticuloendothelial system (RES), non-specific biodistribution, rapid clearance and formation of thrombosis, declining therapeutic efficiency of drug-loaded NPs." needs references
Responses: We have added reference.
The paragraph "From the pharma-83 ceutical point of view, premature degradation before reaching the target site results in 84 the drug's sub-therapeutic action. So, it is necessary to minimize the unwanted clearance 85 of NPs for maximizing their therapeutic potential and targetability, while reducing the 86 off-target effects." needs references.
Responses: We have added the reference.
The paragraph "A surface modification would be a potential approach to avoid opsonization and sys-88 temic instability of NPs for improved therapeutic action and reduced side effects. Poly-89 ethylene glycol (PEG), a water-soluble, non-ionic and biocompatible coating material, is 90 widely used to extend the blood circulation time of drug-loaded NPs and thus enhance the therapeutic efficacy (30-34). PEGylation is the most acceptable way to reduce the 92 protein binding site on NPs, thus inhibiting the macrophage uptake via opsonin recep-93 tors and extending the half-life of the NPs in blood. Moreover, it also increases the size 94 and flexibility of NPs, reducing their renal filtration, and enhancing the extravasation by 95 forming softer NPs (35-38)." presents repeated information. Also the foloowing reference must be added. DOI: 10.1039/c7nr01607g
Responses: We have corrected and cited the articles.
Section 2.7. Stability of PEG-SSNs in mice plasma: Can authors confirm Viscosity and refractive index than they used for these measuremnts? Moreover, authors are measuring diluted plasma, not real samples. Can authors check in non-diluted plasma?
Responses: We have used viscosity 0.8872 of and refractive index of 1.330 for conducting DLS measurement. We took 10 % plasma collected from mice to analyze the stability of formulated NPs in mice plasma. By which we can hypothesize the particle size distribution after systemic administration.
Authors descrbe in section 2.8 siRNA encapsulation efficiency of PEG-SSNs. However, is a physical absorption, not an encapsulation. Please correct it.
Responses: We have corrected.
There is lack of a releasing profile. Authors can check AF 488 fluorescence along the time in PBS or/and plasma.
Responses: We have added the release profile of AF-488 fluorescence siRNA at different pH, which will demonstrate the release mechanism of siRNA from NP/siRNA complexes inside the cell cytoplasm.
If would be nice if authors perform more cytotoxicity test, in addition to hemolysis and MTT. There is lack of description of the positive control in the MTT., ethanol? Triton-X? Moreover, in order to evaluate the side effects of these new nanoparticles, the cytotoxicity over healthy cells is necessary.
Responses: We have added the MTT cytotoxicity results of formulated SSNs and PEG-SSNs over healthy HEK kidney cells.
Section 2.15. The authors do not extract all the blood from the animals before their sacrifice. They also homogenize the tissues before measuring fluorescence. How do you know that the fluorescence comes from particles that are truly accumulated and not from those present in blood? Even more important, why authors do not use AF488 as control? For in vivo experiments, is necesary 2 controls: empty NPS and drug alone.
Responses: We calculated the fluorescence intensity of each organ by subtracting the fluorescence of control tumor bearing mice (no treatment) from treated mice after homogenization. It is hypothesized that the naked siRNAs are rapidly degraded by enzyme RNAses and nucleases present in the blood and that’s why we have not use empty siRNAs to reduce the mice consumption. Moreover, the previous published articles from our lab showed there is no significant difference between the control, free NPs and free siRNAs(Tiash, Kamaruzman et al. 2017). To reduce the number animal consumption, we didn’t use free siRNAs.
In section 2.15, authors describe "When the tumor volume reached 75 mm3 on day 11 of injection", while in section 2.16, authors describe "After the tumor reached an average volume of ~22 mm3 at around day 10-11 of subcuta-311 neous injection". Why did they use a lower size? why did tumors grow slowly? Moreover, the utilization vernier calipers to estimate tumor volume, which lacks accuracy and can lead to overestimation of tumor reduction, that is, underestimation of total tumor volume. Tumor masses are usually infiltrative and often show ingrowth patterns that require the use of medical imaging techniques to accurately determine tumor volume.
Responses: For biodsitribution study and the mice were injected at day 15 when tumor volume reached average 75 mm3. We have corrected this. The tumor volume grow pattern were same for the both study. For tumor volume measurement, if the subcutaneous injection is perfect then there is less chance of ingrowth pattern and are easily measurable with Vernier calipers.
Section 3.1.
TEM images in each step is necessary.
Since all DLS are above 800 nm. How do authors explain the application of these NPs in preclinical models of cancer, since 200 nm have been extensively described as the upper limit?
Responses: To prepare the nanoparticles we have not used any FBS. When we inject the nanoparticles into the system the NPs coat with blood protein that might reduce the particle-particle aggregation. Moreover, from the stability study, we have found that the particle size and distribution are in nanometer range. We assume that in the plasma the particle will be remain in non-aggregated form in systemic circulation
Figure 3. Please show all images with the same magnification.
Responses: We have added the same magnification images
Figure 5. TEM of NPs revealed more than 500 nm, but DLS in blood is around 10. In my experience it is posible only by 2 ways: 1. Digestion Ripenning. 2. A very low concentration of NPs, and thwe measurement of the proteins (with sizes around 10 nm). Please check by TEM and discuss
Responses: We hypothesize that the nanoparticles into the blood circulation will encounter with several blood proteins which might reduce the particle-particle interaction, thereby reducing the particle diameter. From our previous study we have found that the addition of 10% plasma reduced the particle-particle aggregation, resulting in smaller particle size. We also assume that the given NPs suspension would be diluted in the mice blood.
Section 3.6. TEM or ICP of cells is necessary to confirm that NPs have been uptaken, and not the siRNA is released, from NPs, outside of cells and is internalized alone.
Responses: Confocal microscopic analysis clearly demonstrated the uptake pattern of NPs by breast cancer cell lines
Figures 7 and 8 exhibited almost the same information. Please combine them.
Responses: We have added the cytotoxicity results of formulated nanoparticles in healthy HEK cells in figure 07 as per your recommendation.
Since 99% of 99% of strontium in the human body is localized in bones and teeth, the ICP of the tissues help to better understand the biodistribution.
Responses: We have correlated data among the protein corona profiling, biodistribution and tumor regression study. We do believe that the obtained data will help to give an overview of behavior of SSNs or PEG-SSNs both in vitro and in vivo for further future study.
Figure 11c present a low magnification, and the tissues cannot be properly observed. Please correct it.
Responses: We have provided original images at PPT slides to get original magnification.
Reviewer 2 Report
This is an interesting and complete study about PEGylated strontium sulfite nanoparticles improving breast tumor-selective delivery of siRNA. However, several points as below should be solved before it could be considered for publication in JFB.
1. In the Introduction, other polymers modifying NPs to reduce the protein adsorption should be mentioned. Several studies (Biomacromolecules 17 (6), 2010-2018; Molecular pharmaceutics 10.10 (2013): 3892-3902) should be included to support such discussion.
2. The authors may add a discussion about the mechanism of biotin-PEG modifying the nanoparticles.
3. Why was the negatively charged siRNA encapsulated in the negatively charged nanoparticles?
4. The resolution and quality of figures 2, 5, 7, 8, 10, and 11 should be improved to a higher level.
5. The cell viability of the nanoparticles is not great (80%-90% under experimental conditions).
6. Technical issues. Line 140, 'SrCl2', 'Na2SO3'; line 141, '370c'; line 221, the equation. Please check all.
Author Response
Reviewer #2:
Comments and Suggestions for Authors
This is an interesting and complete study about PEGylated strontium sulfite nanoparticles improving breast tumor-selective delivery of siRNA. However, several points as below should be solved before it could be considered for publication in JFB.
01.In the Introduction, other polymers modifying NPs to reduce the protein adsorption should be mentioned. Several studies (Biomacromolecules 17 (6), 2010-2018; Molecular pharmaceutics 10.10 (2013): 3892-3902) should be included to support such discussion.
Responses: We have discussed and added reference.
- The authors may add a discussion about the mechanism of biotin-PEG modifying the nanoparticles.
Responses: We have discussed and added reference
- Why was the negatively charged siRNA encapsulated in the negatively charged nanoparticles?
Responses: Although the net charge of our developed nanoparticles was negative, they actually contain both negative and positive charges responsible for encapsulating the siRNAs through electrostatic interaction.
- The resolution and quality of figures 2, 5, 7, 8, 10, and 11 should be improved to a higher level.
Responses: We provided the original images through PPT for improved resolution.
- The cell viability of the nanoparticles is not great (80%-90% under experimental conditions).
Responses: Actually, the formulated SSNs particle tend to sediment quickly which will cause cell wall rupture. This is why the nanoparticles shows cytotoxicity. From the in vivo toxicity study we found these two formulations are safe and nontoxic.
- Technical issues. Line 140, 'SrCl2', 'Na2SO3'; line 141, '370c'; line 221, the equation. Please check all.
Responses: We have corrected
Reviewer 3 Report
The results of this paper are interesting but lack some basic evidence to ensure reliability. This reviewer strongly recommends that the authors resubmit the paper with sufficient supplemental experiments, regardless of the revision deadline.
(1) Biotin binds to streptavidin with high affinity and has been used to link various compounds in biological applications. However, from reading the Materials and Method section, it appears that streptavidin is not loaded on SSN, but passively adsorbed to SSN surface, how stable can it be immobilized on the particles? By what binding mode can biotin-PEG be immobilized on the particles? In addition, if it is in the absence of streptavidin in the first place, the significance of the presence of biotin wit PEG is unclear.
(2) The decrease in turbidity with the addition of PEG shown in Figure 1 is likely due to the progression of aggregation and a decrease in the number of particles. From this turbidity data, it is difficult to understand the relevance of "slowing down the rate of particle formation" that the authors claim. If the results were to show changes in turbidity over time, they might be meaningful, but showing a comparison of the turbidity of samples at a given point in time does not seem to make much sense.
(3) The dynamic scattering data shown in Figure 2(C) do not match the description and the results, and there does not seem to be much difference between the data with and without PEG.
(4) The SEM data shown in Figure 3 lacks a quantitative argument. The morphological changes claimed by the authors seem too subjective.
(5) Would it be possible to calculate the content of PEG introduced into the particles from the EDX results in Figure 4? Also, is that content consistent with data from other experiments?
(6) Is it possible to obtain a micrograph or SEM image of PEG-SSN treated with the blood shown in Figure 5(A)? We need more reliable evidence that changes are really taking place in the blood that reduce the diameter by two orders of magnitude.
(7) What exactly is the amount of siRNA loading shown in Figure 5(b)?
(8) A major problem with this paper is that it has not been proven whether RNA interference is occurring with RNA-loaded PEG-SSN. I strongly recommend that the cancer tissue shown in Figure 10(B) should be confirmed by quantitative PCR to reliably show that siRNA reduces the amount of EGFR mRNA in the cells.
Author Response
Reviewer #3:
Comments and Suggestions for Authors
The results of this paper are interesting but lack some basic evidence to ensure reliability. This reviewer strongly recommends that the authors resubmit the paper with sufficient supplemental experiments, regardless of the revision deadline.
01.Biotin binds to streptavidin with high affinity and has been used to link various compounds in biological applications. However, from reading the Materials and Method section, it appears that streptavidin is not loaded on SSN, but passively adsorbed to SSN surface, how stable can it be immobilized on the particles? By what binding mode can biotin-PEG be immobilized on the particles? In addition, if it is in the absence of streptavidin in the first place, the significance of the presence of biotin wit PEG is unclear.
Responses: We presented in our earlier publication that while streptavidin helps link biotin-PEG to nanoparticles, biotin itself through its protonated amine groups can significantly assist in linking PEG to the nanoparticles.
Ref: Mozar FS, Ezharul Hoque Chowdhury EH.PEGylation of Carbonate Apatite Nanoparticles Prevents Opsonin Binding and Enhances Tumor Accumulation of Gemcitabine. J. Pharm Sci. 2018 Sep;107(9):2497-2508. doi: 10.1016/j.xphs.2018.05.020
02.The decrease in turbidity with the addition of PEG shown in Figure 1 is likely due to the progression of aggregation and a decrease in the number of particles. From this turbidity data, it is difficult to understand the relevance of "slowing down the rate of particle formation" that the authors claim. If the results were to show changes in turbidity over time, they might be meaningful, but showing a comparison of the turbidity of samples at a given point in time does not seem to make much sense.
Responses: From our previous published studies, we found that the decrease of turbidity is due to less aggregation of the particles, resulting in the smaller particle size. As we know, PEG helps to keep the nanoformulations in colloidal state and prevent particle-particle aggregation. The changes of particle agglomeration were confirmed by (optical) microscopic observation of NPs.
03.The dynamic scattering data shown in Figure 2(C) do not match the description and the results, and there does not seem to be much difference between the data with and without PEG.
Responses: The dynamic scattering of SSNs within the range of 1000-10000nm whereas PEG-SSNs were below 1000nm range. The particle size of plain SSNs was around 2000nm. On the other hand, the PEG-SSNs had the particle size in the range of 600-800 nm.
04.The SEM data shown in Figure 3 lacks a quantitative argument. The morphological changes claimed by the authors seem too subjective.
Responses: We have added the images in the same magnification to compare the morphology. The only way to evaluate the difference is visualization through SEM. We have found the evidence of reduction of particle size of PEG-SSNs from SEM study maintaining the similar size reduction pattern observed through DLS.
05.Would it be possible to calculate the content of PEG introduced into the particles from the EDX results in Figure 4? Also, is that content consistent with data from other experiments?
Responses: The results showed the significant difference between SSNs with or without PEG in the percent weight of C and O in the formulation. We have found the increase ratio of Carbon and Oxygen after adding PEG, which we assume the effect of PEG grafting onto the SSNs. Yes, we are confident this content will be consistent in the other experiments.
- Is it possible to obtain a micrograph or SEM image of PEG-SSN treated with the blood shown in Figure 5(A)? We need more reliable evidence that changes are really taking place in the blood that reduce the diameter by two orders of magnitude.
Responses: From our previous experience when we added the 10% of FBS the size of particles reduced significantly. Incubation of particles in blood assumed to be coated with blood protein around the surface of SSNs which might prevent the particle agglomeration resulted in notable reduction of diameter of particles
07.What exactly is the amount of siRNA loading shown in Figure 5(b)?
Responses: We have not calculated the amount but from the binding affinity study we have found that both SSNs and PEG-SSNs showed more than 80% of binding affinity towards the siRNAs and also showed significant cellular internalization found in confocal microscopic analysis.
08.A major problem with this paper is that it has not been proven whether RNA interference is occurring with RNA-loaded PEG-SSN. I strongly recommend that the cancer tissue shown in Figure 10(B) should be confirmed by quantitative PCR to reliably show that siRNA reduces the amount of EGFR mRNA in the cells.
Responses: We have added the expression level of EGFR in MCF-7 cells with or with treatment of EGFR siRNA/ SSNs complexes.
Reviewer 4 Report
The manuscript contains lots of experiments and results. However, the organization and writing have to be sufficiently improved. There could be more revision needed after primary correction. The selective comments for initial revision are:
[1] The authors need to care about the length of sentences, especially in the abstract and discussion. It would be better splitting into two sentences of one long line. Then the reader will get in touch with reading.
[2] In figure 2, the authors could use the lowercase alphabet when it is repeated (ABC) or try to use a different symbol. Figure (2C) also shows raw (images) data collected from the zetasizer software. The authors need to plot this data and the same pattern to compare size distribution analysis. They can choose a sigma plot or other convenient tools.
[3] Figure 3 (FE-SEM), should be the same scale bar (500 nm or 1μm or 5μm). Otherwise, it is very difficult to compare size and morphology.
[4] The authors could merge Figures 4(B) and 4(C); Figures 4E and (4F). It will help the data organization.
[5] Figure 5 needs to plot again so that the x-axis and y-axis are readable.
[6] The Figure 6 images need a scale bar.
[7] The discussion section should be based on reported results. Unnecessary discussion may confuse the reader. What is not part of this study does not need to mention in the discussion part.
Author Response
Reviewer #4:
Comments and Suggestions for Authors
The manuscript contains lots of experiments and results. However, the organization and writing have to be sufficiently improved. There could be more revision needed after primary correction. The selective comments for initial revision are:
[1] The authors need to care about the length of sentences, especially in the abstract and discussion. It would be better splitting into two sentences of one long line. Then the reader will get in touch with reading.
Responses: We have corrected and split the sentences.
[2] In figure 2, the authors could use the lowercase alphabet when it is repeated (ABC) or try to use a different symbol. Figure (2C) also shows raw (images) data collected from the zetasizer software. The authors need to plot this data and the same pattern to compare size distribution analysis. They can choose a sigma plot or other convenient tools.
Responses: We have corrected. We think the raw data from zetasizer are very suitable for comparison of particle size distribution.
[3] Figure 3 (FE-SEM), should be the same scale bar (500 nm or 1μm or 5μm). Otherwise, it is very difficult to compare size and morphology.
Responses: We have corrected.
[4] The authors could merge Figures 4(B) and 4(C); Figures 4E and (4F). It will help the data organization.
Responses: We have rearranged the images.
[5] Figure 5 needs to plot again so that the x-axis and y-axis are readable.
Responses: We provided the original PPT images.
[6] The Figure 6 images need a scale bar.
Responses: We have added scale bar.
[7] The discussion section should be based on reported results. Unnecessary discussion may confuse the reader. What is not part of this study does not need to mention in the discussion part.
Responses: We have modified
Round 2
Reviewer 1 Report
Authors has solved all reviewer questions. Therefore, the work can be published in the present form.
Author Response
Thank you very much.
Reviewer 2 Report
accept
Author Response
Thank you very much.
Reviewer 3 Report
"01.Biotin binds to streptavidin with high affinity and has been used to link various compounds in biological applications. However, from reading the Materials and Method section, it appears that streptavidin is not loaded on SSN, but passively adsorbed to SSN surface, how stable can it be immobilized on the particles? By what binding mode can biotin-PEG be immobilized on the particles? In addition, if it is in the absence of streptavidin in the first place, the significance of the presence of biotin wit PEG is unclear.
Your Responses: We presented in our earlier publication that while streptavidin helps link biotin-PEG to nanoparticles, biotin itself through its protonated amine groups can significantly assist in linking PEG to the nanoparticles. Ref: doi: 10.1016/j.xphs.2018.05.020"
Reviewer's Comment
Since the amines in biotin are in cyclic ureides, does protonation really occur? If protonation does occur under special conditions, please provide the pKa value. Also, the prior experiment you cited was an adsorption reaction on carbonate apatite nanoparticles, which is different from the material used in this experiment, so I think this is an insufficient response to this reviewing point.
"03.The dynamic scattering data shown in Figure 2(C) do not match the description and the results, and there does not seem to be much difference between the data with and without PEG.
Your Responses: The dynamic scattering of SSNs within the range of 1000-10000nm whereas PEG-SSNs were below 1000nm range. The particle size of plain SSNs was around 2000nm. On the other hand, the PEG-SSNs had the particle size in the range of 600-800 nm."
Reviewer's Comment
It is difficult to recognize them, so why don't you make a more enlarged paricle size distribution diagram of DNS?
Author Response
Reviewer #3:
01.Biotin binds to streptavidin with high affinity and has been used to link various compounds in biological applications. However, from reading the Materials and Method section, it appears that streptavidin is not loaded on SSN, but passively adsorbed to SSN surface, how stable can it be immobilized on the particles? By what binding mode can biotin-PEG be immobilized on the particles? In addition, if it is in the absence of streptavidin in the first place, the significance of the presence of biotin wit PEG is unclear.
Responses: We presented in our earlier publication that while streptavidin helps link biotin-PEG to nanoparticles, biotin itself through its protonated amine groups can significantly assist in linking PEG to the nanoparticles. Ref: doi: 10.1016/j.xphs.2018.05.020"
Reviewer's Comment
Since the amines in biotin are in cyclic ureides, does protonation really occur? If protonation does occur under special conditions, please provide the pKa value. Also, the prior experiment you cited was an adsorption reaction on carbonate apatite nanoparticles, which is different from the material used in this experiment, so I think this is an insufficient response to this reviewing point.
Responses: Yes, protonation is supposed to be occur. At a pH greater than 10 (isoelectric point), the amine exists as a neutral base, which means that amine groups in biotin should be protonated at pH 7.5 of the culture medium which was added to the formulated particles. Similar to the carbonate apatite, strontium sulfite has the cationic domains (Sr++) and anionic domains SO32- to form the nanocomplexes with the chemicals or drugs or nucleic acids through electrostatic interactions.
"03. The dynamic scattering data shown in Figure 2(C) do not match the description and the results, and there does not seem to be much difference between the data with and without PEG.
Your Responses: The dynamic scattering of SSNs within the range of 1000-10000nm whereas PEG-SSNs were below 1000nm range. The particle size of plain SSNs was around 2000nm. On the other hand, the PEG-SSNs had the particle size in the range of 600-800 nm.
It is difficult to recognize them, so why don't you make a more enlarged particle size distribution diagram of DNS?
Responses: we have enlarged the diagram.
Reviewer 4 Report
The authors have done most of the corrections. Some minor correction needs to address:
[1] The Figure 2C (i-iv) description is missing in the figure description. The Figure 2D, E, F are absences in figure.
[2] The Figure 12B images need a scale bar.
[3] There are some typo and symbol error that should be corrected in revision submission.
Author Response
Reviewer #4:
Comments and Suggestions for Authors
The authors have done most of the corrections. Some minor correction needs to address:
[1] The Figure 2C (i-iv) description is missing in the figure description. The Figure 2D, E, F are absences in figure.
Responses: We have corrected
[2] The Figure 12B images need a scale bar.
Responses: We have added scale bar.
[3] There are some typo and symbol error that should be corrected in revision submission.
Responses: We have corrected.